# The Effect of Plant Growth Promoting Rhizobacteria *Bacillus thuringiensis* LKT25 on Cadmium Accumulation and Physiological Responses in *Solanum nigrum* L.

**DOI:** 10.3390/plants14182918

**Published:** 2025-09-19

**Authors:** Guannan Kong, Da Song, Chao Zhang, Xinyao Jia, Yingying Ren, Shuhe Wei, Huiping Dai

**Affiliations:** 1School of Biological Science and Engineering, Shaanxi University of Technology, Hanzhong 723000, China; konggn@snut.edu.cn (G.K.); songda@snut.edu.cn (D.S.); zc13572734625@163.com (C.Z.); 18966711062@163.com (X.J.); 18089151006@163.com (Y.R.); 2Key Laboratory of Pollution Ecology and Environment Engineering, Institute of Applied Ecology, Chinese Academy of Sciences, Shenyang 110016, China; shuhewei@iae.ac.cn

**Keywords:** heavy metal (Cd), *Solanum nigrum* L., super accumulator plant, plant growth-promoting rhizobacteria, *Bacillus thuringiensis* LKT25, antioxidation system

## Abstract

Cadmium contamination in soil threatens ecological safety and human health. Phytoremediation has gained attention due to its cost-effectiveness and environmental sustainability. Studies show that plant growth-promoting rhizobacteria can enhance the ability of hyperaccumulator plants to remove heavy metals. This research aimed to isolate and identify plant-growth-promoting rhizobacteria under Cd stress and assess their impact on the growth and Cd accumulation of *Solanum nigrum* L. Six bacterial strains were isolated from the rhizosphere of *S. nigrum*, all showing high Cd tolerance. Among them, LKT25 exhibited multiple growth-promoting traits, including indole-3-acetic acid production, nitrogen fixation, 1-aminocyclopropane-1-carboxylate deaminase, and siderophore synthesis. Under varying Cd concentrations (5, 25, and 50 mg/kg), the *Bacillus thuringiensis* strain LKT25 significantly improved Cd removal by *S. nigrum*. At 5 mg/kg Cd, the removal efficiency reached 45.13%. LKT25 also enhanced plant growth, photosynthesis, and antioxidant activity, contributing to improved Cd remediation. This study provides new microbial resources and technical support for using rhizobacteria in remediating heavy metal-contaminated soils.

## 1. Introduction

With the intensification of human activities and the acceleration of industrialization, heavy metal pollution in soil environments has increasingly threatened human living conditions [1,2,3]. Cadmium (Cd) recognized as one of the most toxic heavy metal pollutants, has emerged as a critical contaminant in agricultural soils, especially in paddy fields [4,5]. Its accumulation in crops and subsequent transfer into the food chain pose significant risks to human health, thereby drawing extensive attention from both the scientific community and the general public [6,7,8]. At present, Cd pollution in China represents the most severe form of heavy metal contamination, with 7.0% of monitored sites exceeding the threshold outlined in the Soil Environmental Quality Standard (GB15618-1995) issued by the Ministry of Ecology and Environment of the People’s Republic of China under pH conditions of 6.5–7.5 (0.6 mg/kg) [9]. Notably, Cd concentrations exceeding 3 mg/kg surpass the risk control threshold, significantly elevating the likelihood of contamination in agricultural products (GB15618-1995). Particularly alarming is the situation in mining areas, where Cd levels have been recorded at over 80 mg/kg [10]. The remediation of Cd in agricultural soils constitutes a critical environmental challenge that demands urgent and effective intervention strategies [10]. A variety of remediation techniques have been proposed to address soil heavy metal pollution. However, conventional physical and chemical approaches are often constrained by high operational costs and the potential to disrupt soil microbial ecosystems [11]. Recent studies have demonstrated that heavy metal stress can trigger interactive mechanisms between plants and microorganisms, enabling them to collaboratively mitigate metal toxicity [12,13,14]. Based on these findings, plant-microbe-based remediation strategies have been developed, which are characterized by their environmental sustainability, high remediation efficiency, cost-effectiveness, and operational simplicity, making them promising solutions for the long-term management of contaminated soils [5,15].

Hyperaccumulator plants, such as *Solanum nigrum* L., which is defined as a plant capable of accumulating more than 100 mg/kg Cd in its above-ground tissues under natural conditions, while maintaining a translocation factor (shoot-to-root ratio) greater than 1 [16]. However, different ecotypes of *S. nigrum* demonstrate considerable differences in their capacity to accumulate Cd under soil cultivation conditions. For example, the ecotype originating from the mountainous regions of Linhai City, Zhejiang Province, China, exhibited root and stem Cd concentrations ranging from 8.93 to 58.15 mg/kg when exposed to soil Cd concentrations of 25–100 mg/kg [17]. In contrast, the population transplanted from Lizi Park in Nanshan District, Shenzhen City, Guangdong Province, displayed root and stem Cd concentrations of 122.7 mg/kg and 82.6 mg/kg, respectively, under a soil Cd concentration of 50 mg/kg [17]. Moreover, the Korean ecotype collected from Daegu, South Korea, showed Cd concentrations in roots, stems, and leaves ranging from 120.49 to 3162.83 mg/kg under soil Cd levels of 10–80 mg/kg [18]. Its practical application is constrained by several critical factors, including the inherently low biomass of plants, the prolonged remediation period, and the oxidative damage induced by high concentrations of Cd, which ultimately inhibits plant growth [19,20]. To counteract such stress, plants possess a sophisticated defense system composed of both enzymatic and non-enzymatic antioxidants that protect cellular structures from oxidative damage. The enzymatic components include superoxide dismutase (SOD), peroxidase (POD), catalase (CAT), and ascorbate peroxidase (APX), whereas the non-enzymatic components consist of key antioxidants such as glutathione and ascorbic acid. Collectively, these mechanisms form a robust defense system that enables plants to withstand adverse environmental stresses [21,22,23].

Plant growth-promoting rhizobacteria (PGPR) is a collective term for a group of bacteria that colonize the root surfaces and thrive in the rhizospheric microenvironment of plants. These bacteria are capable of significantly enhancing plant growth and improving the efficiency of phytoremediation through both direct and indirect mechanisms, including the alleviation of heavy metal toxicity [2]. Direct mechanisms mainly involve the secretion of siderophores, organic acids, surfactants, nitrogenase, 1-aminocyclopropane-1-carboxylate (ACC) deaminase, and indole-3-acetic acid (IAA) plant growth hormones by PGPR, which collectively promote plant growth, enhance biomass accumulation under heavy metal stress conditions, and thereby improve the overall efficiency of plant-based remediation [24]. Indirect mechanisms include the activation of plant defense systems and the reduction in soil heavy metal ion mobility and bioavailability, which together contribute to increased plant tolerance to heavy metal stress [25,26].

Among PGPR, strains of the genera *Pseudomonas* and *Bacillus* have been extensively studied due to their strong environmental adaptability and diverse plant growth-promoting traits [2,27]. However, their synergistic remediation mechanisms in association with *S. nigrum* remain insufficiently understood and warrant further investigation.

Previous studies have demonstrated that PGPR can enhance the growth of hyperaccumulator plants [2]. However, most studies to date have focused primarily on the effects of limited promoting factors under single Cd concentration conditions. In contrast, research on PGPR with multiple functions, especially their roles in improving plant physiological traits and soil Cd remediation across varying contamination levels, remain limited. Furthermore, although *Bacillus thuringiensis* is widely utilized in agriculture as a biopesticide [28], its potential in promoting plant growth and Cd-tolerance, particularly in regulating Cd accumulation and physiological responses in *S. nigrum*, remains largely underexplored. In this context, six Cd-tolerant bacterial strains were isolated from the rhizosphere of *S. nigrum*. A novel Cd-tolerant functional strain, *Bacillus thuringiensis* LKT25, exhibiting multiple plant growth-promoting traits, was identified through comprehensive physiological, biochemical, and molecular characterization. The differential effects of strain on plant growth, physiological responses, and Cd accumulation under varying Cd stress levels, as well as its underlying mechanisms, were systematically investigated. This research provides a theoretical foundation for the application of multi-functional Cd-tolerant microorganisms in enhancing the stability and efficiency of phytoremediation under moderately to highly polluted and complex environmental conditions, and contributes efficient and high-quality microbial resources for the development of sustainable heavy metal pollution remediation technologies.

## 2. Results

### 2.1. Molecular Identification and Cd Tolerance of Six Rhizosphere Bacterial Strains and Assessment of Their Plant Growth-Promoting Properties

Through 16S rRNA sequencing and phylogenetic analysis, six bacterial isolates were classified into three distinct genera: *Bacillus* (strains LKT17 and LKT25), *Pseudomonas* (strains Y80, Y89, and S36), and *Prizedia* (strain LKT38) (Figure 1).

All six isolates demonstrated the capacity to produce IAA, as confirmed by qualitative assays. Quantitative analysis (Figure 2) further revealed that IAA production levels varied among the strains, ranging from 4.24 to 46.03 mg/L. Among them, strain LKT25 exhibiting the highest IAA yield, reaching 46.03 mg/L. Additionally, all six strains displayed the ability to synthesize nitrogenase (Table 1 and Figure A1) and siderophores (Table 1 and Figure A2). Notably, only strain LKT25 has the ability to utilize ACC (Table 1 and Figure A3), a trait associated with plant stress mitigation.

Cd tolerance assays (Figure 3) showed that five strains (LKT17, LKT25, LKT38, Y80, and Y89) maintained consistent growth rates in the presence of 50 mg/L Cd, indicating a high level of metal tolerance. Given its comprehensive suite of plant growth-promoting characteristics and robust Cd resistance, strain LKT25 was selected for further investigation in combination with the hyperaccumulator plant *S. nigrum* to assess its impact on Cd uptake across varying contamination levels.

### 2.2. The Effect of LKT25 on the Growth and Development of S. nigrum Under Different Concentrations of Cd Stress

This study was designed to assess the impact of LKT25 on key growth and development parameters of *S. nigrum* under varying soil Cd concentrations (0, 5, 25, and 50 mg/kg), including root length, plant height, and biomass accumulation. Experimental results revealed that LKT25 application significantly stimulated root elongation across all Cd treatments (Figure 4a). Particularly at a Cd concentration of 5 mg/kg, the enhancement in root length was most significant, reaching 2.6-fold higher than that of the control group (no LKT25 treatment). A similar trend was observed for plant height (Figure 4b), where the maximum increase 1.92-fold higher than that of the control was recorded under the same 5 mg/kg Cd condition. Further analysis demonstrated that, under this specific Cd level, both root length and plant height were restored to levels comparable to those of plants cultivated in Cd-free soil. Moreover, biomass measurements showed that LKT25 significantly enhanced the dry weight of both shoot and root (shoot: 2.87- to 6.6-fold, root: 1.89- to 2.68-fold) components of plants across all tested Cd concentrations (Figure 4c,d).

### 2.3. The Effect of LKT25 on Cd Accumulation of the S. nigrum

This study further investigated the impact of LKT25 application on Cd accumulation in *S. nigrum* under different soil Cd concentration (0, 5, 25, and 50 mg/kg). The results demonstrated that LKT25 significantly enhanced Cd accumulation in the shoot at a soil Cd concentration of 5 mg/kg, reaching 2.07-foldthat of the control group (Figure 5a). At 25 mg/kg, no statistically significant difference was observed between the LKT25-treated and untreated groups. However, at 50 mg/kg, Cd content in the shoot decreased by 34% compared to the control (Figure 5a). Furthermore, the application of LKT25 significantly increased the Cd content in the roots of plants under soil Cd concentrations of 25 and 50 mg/kg, with values amounting to 1.32 and 1.72-fold those of the control group, respectively. In contrast, no statistically significant difference was detected at a soil Cd concentration of 5 mg/kg when compared to the control group (Figure 5b).

The enrichment coefficient is defined as the ratio of heavy metal concentrations in the shoots and roots of a plant to the corresponding concentrations in the soil, and it serves as an indicator of the plant’s capacity to accumulate heavy metals from contaminated soils. In this study, the enrichment coefficients of both the LKT25 treatment group and the control group across all tested soil Cd concentrations were markedly higher than 1 (Table 2). Compared with the control group, LKT25 application significantly increased the enrichment coefficients in the shoots of *S. nigrum* at soil Cd concentrations of 5 and 50 mg/kg, as well as in the root parts at concentrations of 25 and 50 mg/kg (Table 2).

The translocation coefficient is defined as the ratio of heavy metal concentrations in the shoots of a plant to those in the roots, serving as an indicator of the efficiency with which plants translocate heavy metals from root to shoot. In this study, the transfer coefficient of *S. nigrum* was observed to exceed 1 exclusively under soil Cd concentrations of 5 mg/kg (Table 2).

The Cd removal rate is defined as the ratio of soil Cd content before and after the cultivation of hyperaccumulator plants under varying soil Cd concentrations, and it serves as an indicator of the overall effectiveness of plant-based remediation in Cd-contaminated soils. In this study, the Cd removal efficiency of *S. nigrum* reached a maximum of 31.12% at a soil Cd concentration of 5 mg/kg, which further increased to 45.13% following the application of LKT25. At 25 mg/kg, the removal efficiency was 15.32%, and LKT25 application resulted in an increase to 22.95%. At 50 mg/kg, the removal efficiency was initially 21.75%, and it rose to 27.86% after LKT25 treatment. The aforementioned research findings demonstrate that, at a soil Cd concentration of 5 mg/kg, the application of LKT25 achieves the highest efficiency (45.13%) of Cd remediation by *S. nigrum* in contaminated soils.

### 2.4. The Effect of LKT25 on Photosynthetic Parameters of the S. nigrum

This study systematically evaluated the influence of LKT25 on chlorophyll content in *S. nigrum* under a range of soil Cd concentration (0, 5, 25, and 50 mg/kg), as illustrated in Figure 6a. Exposure to Cd significantly suppresses chlorophyll synthesis. The application of LKT25 resulted in a marked increase in chlorophyll content across all Cd treatments. Particularly, at a Cd concentration of 5 mg/kg, chlorophyll levels were restored to levels comparable to those of the control group under 0 mg/kg Cd conditions, indicating that LKT25 can effectively counteract the phytotoxic effects of Cd in *S. nigrum*.

The Cd accumulation capacity of hyperaccumulator plants is closely correlated with their biomass, as the photosynthetic rate represents a fundamental energy source driving plant growth. This study systematically assessed the impact of LKT25 on the photosynthetic performance of *S. nigrum* under varying soil Cd concentration (0, 5, 25, and 50 mg/kg), as illustrated in Figure 6b. Exposure to Cd significantly reduces the photosynthetic rate in *S. nigrum*, with the inhibitory effect intensifying as Cd concentration increased. The application of LKT25 significantly enhanced photosynthetic efficiency across all tested Cd concentrations. Notably, at a soil Cd concentration of 5 mg/kg, chlorophyll content was restored to a level comparable to those of the control group under 0 mg/kg Cd conditions.

### 2.5. The Effect of LKT25 on the Antioxidant Enzyme Activity of S. nigrum

In this study, the SOD activity in both roots and shoots of *S. nigrum* decreased with increasing Cd concentration (Figure 7a,b). Inoculation with LKT25 significantly elevated SOD activity in the shoots at Cd concentrations of 5 and 25 mg/kg, as well as in the roots at 5, 25, and 50 mg/kg (Figure 7a). Similarly, POD activity in both the roots and shoots of *S. nigrum* declined with increasing Cd concentration (Figure 7c,d). The application of LKT25 significantly elevated POD activity in the shoots at Cd concentrations of 0, 5, and 25 mg/kg, as well as in the roots across all tested Cd concentrations (Figure 7c,d). CAT levels in both the roots and shoots of *S. nigrum* decreased progressively with increasing Cd concentrations (Figure 7c,d). The application of LKT25 was found to significantly elevate CAT content in both roots and shoots of *S. nigrum* under all tested Cd concentration conditions (Figure 7e,f). APX activity in both the roots and shoots of *S. nigrum* was found to decline with increasing Cd concentration (Figure 7g,h). The application of LKT25 significantly elevated APX activity in both the shoots and roots across all tested Cd concentrations (Figure 7g,h).

The above research findings provide evidence that LKT25 modulates the plant *S. nigrum* antioxidant system through the regulation of key antioxidant enzyme activities, including SOD, POD, CAT and APX, thereby effectively preventing or alleviating oxidative stress-induced damage in plants.

### 2.6. The Effects of LKT25 on Malondialdehyde (MDA) Levels in the Plant S. nigrum

In this study, the MDA content in both the aerial parts and roots of *S. nigrum* plants cultivated under increasing soil Cd concentrations (0, 5, 25, and 50 mg/kg) exhibited a progressive and dose-dependent increase (Figure 8a,b). Application of LKT25 significantly reduced MDA accumulation in both plant tissues at Cd concentrations of 5, 25, and 50 mg/kg. These results demonstrate that LKT25 application effectively alleviates oxidative stress and protects plant cell membranes from Cd-induced damage, particularly under moderate stress conditions.

## 3. Discussion

### 3.1. LKT25 Enhances the Growth and Photosynthetic Efficiency of S. nigrum

PGPR significantly enhance plant growth performance by facilitating nutrient uptake and suppressing the proliferation of pathogenic bacteria and pests [29,30]. In this study, bacterial strain LKT25 was identified as *Bacillus thuringiensis* based on 16S rRNA gene sequence analysis. The sequence exhibited a similarity of 98.8% to that of the reference strain Bacillus thuringiensis ATCC 10792 (Figure 1), indicating that strain LKT25 represents a novel strain not previously reported in the literature. Excessive accumulation of heavy metals can exert detrimental effects on plant physiology, either directly or indirectly, leading to growth inhibition [31,32]. *Bacillus* has great potential in the production of auxins, lytic enzymes, siderophores and nitrogen fixation and these growth-promoting factors can enhance plant growth and increase plant’s stress resistance through direct or indirect means [33,34,35]. Among these factors, IAA, at optimal concentrations, promotes the development of xylem and root systems, stimulates the synthesis of photosynthetic pigments, and enhances photosynthetic efficiency [36,37]. Additionally, the synthesis of ACC deaminase synergizes with IAA to facilitate root elongation under short-term conditions [33]. Nitrogen-fixing PGPR capable of biological nitrogen fixation can supply essential nitrogen nutrients to plants, thereby promoting plant growth [33]. In this study, strain LKT25 not only exhibits a high capacity for IAA production (as shown in Figure 2), but also possesses the abilities to utilize ACC (Table 1 and Figure A3) and to perform nitrogen fixation (Table 1 and Figure A1). The IAA production level of this strain reached 46.03 mg/L, which was significantly higher than that of other Cd-tolerant strains analyzed in this research (as shown in Figure 2) and also exceeded the previously reported IAA production range for other strains within the same genus (5.7–22.9 mg/L) [29,38]. Furthermore, LKT25 significantly stimulated root development and biomass in *S. nigrum* (Figure 4a,c,d), with the most pronounced effects observed under a soil Cd concentration of 5 mg/kg. Under this condition, root length increased by 1.75-fold and biomass by 2.69-fold compared to the control group, while the shoot biomass reached 6.6-fold that of the control group, surpassing the levels reported for other species within the *Bacillus* genus [39] and a variety of other bacteria [27,40,41]. Additionally, the application of LKT25 significantly increased the total chlorophyll content and improved photosynthetic efficiency in *S. nigrum* (Figure 6a,b). These findings align with previous studies [29,33,34,35,36,37,38,39], showing that LKT25 significantly reduces the inhibitory effects of Cd stress on photosynthesis and promotes plant growth.

### 3.2. LKT25 Enhances the Antioxidant System of S. nigrum and Reduces Cd-Induced Oxidative Stress

Plant defense against heavy metal stress depends on their ability to maintain redox homeostasis in the antioxidant system [42,43]. Heavy metal exposure triggers ROS overproduction, leading to lipid peroxidation and the formation of cytotoxic MDA [44]. Our results show that LKT25 significantly reduces MDA levels in *S. nigrum* under all tested Cd concentrations (Figure 8a,b), indicating that co-cultivation with LKT25 effectively alleviates oxidative stress.

Plant-microbe co-cultivation mainly enhances antioxidant capacity by regulating antioxidant enzymes and non-enzymatic substances [45,46]. Enzymes like SOD, POD, CAT, and APX help maintain protein stability, preserve membrane integrity, and reduce lipid peroxidation [47]. Our results show that LKT25 significantly boosts antioxidant enzyme activity (SOD, POD, CAT, APX) in *S. nigrum* roots at all tested Cd levels (Figure 7b,d,f,h), reducing oxidative stress. At 0, 5, 25mg/kg tested Cd concentrations, LKT25 also enhances these enzymes in the shoots (Figure 7a,c,e,g). However, at 50 mg/kg, no significant difference in stem SOD and POD activity is observed between treated and control plants (Figure 7b,d). This observation may be attributed to reduced Cd accumulation in the shoots of LKT25-treated plants (177.6 mg/kg) as compared to the control group (269.8 mg/kg, Figure 5a,b). This pattern suggests that LKT25 mainly functions within the roots, potentially by restricting the upward translocation of Cd, resulting in no strong stress response in the stems.

### 3.3. LKT25 Regulates the Accumulation and Transport of Cd in S. nigrum

Hyperaccumulator plants are key to bioremediation because they can greatly reduce heavy metal levels in soil through excessive accumulation [16]. However, high pollution levels often limit their effectiveness [48]. PGPR have been shown to promote plant growth and enhance Cd absorption [27,49]. Microbial-plant co-cultivation systems employ diverse strategies to reduce Cd stress depending on pollution levels [50]. Under low contamination, plant growth-promoting bacteria help plants absorb and transport heavy metals to aboveground parts through processes like IAA secretion, nitrogen fixation, ACC deaminase production, siderophore synthesis [51,52], and organic acid release [33,34,35,36,37]. In high contamination, Cd mainly accumulates in roots (root > stem > leaf), with limited movement to shoots, reducing toxicity [53,54]. In this case, plant-microbe interactions regulate the production of compounds like ACC deaminase, antioxidant enzymes, and non-enzymatic antioxidants to reduce oxidative damage [47,54,55]. This study shows that LKT25 significantly improves the Cd removal ability of *S. nigrum* across pollution levels. At 5 mg/kg soil Cd, LKT25 achieved a 45.13% removal rate (Figure 5a,b), matching current rhizobacteria-assisted phytoremediation methods [38,39,41,56,57]. It also increased shoot Cd by 2.07-fold compared to the control, with a transfer coefficient over 1-higher than other *Bacillus* strains [39,54,55]. At this stage, a favorable interaction between LKT25 and *S. nigrum* facilitated the absorption and upward translocation of Cd, indicating its significant potential for practical application. At a Cd concentration of 25 mg/kg, *S. nigrum* restricts Cd translocation from roots to shoots via intrinsic regulatory mechanisms, resulting in a transport coefficient below 1. This restriction corresponded to the lowest Cd removal efficiency observed, which was lower than that at both 5 mg/kg and 50 mg/kg. Applying LKT25 significantly increased root Cd accumulation and elevated oxidative stress levels, effectively protecting aboveground tissues and improving overall Cd removal. In contrast, under 50 mg/kg Cd stress, *S. nigrum* absorbed more Cd than at 25 mg/kg, suggesting strong activation of defense and tolerance mechanisms. This stress may up-regulate genes involved in heavy metal absorption, translocation, and compartmentalization [58,59], promoting Cd movement to shoots, with a transport coefficient of 1.12-higher than the 1.05 observed at 5 mg/kg. Under this condition, LKT25 further increased root Cd enrichment and reduced shoot translocation (coefficient < 1), alleviating plant stress. These results show that LKT25 not only protects plants from Cd-induced damage but also has strong synergistic potential in high-Cd environments, with possible effectiveness at concentrations above 50 mg/kg.

## 4. Materials and Methods

### 4.1. Isolation and Identification of PGPR from S. nigrum

Rhizosphere soil samples were collected from the roots of *S. nigrum* plants growing in farmland located in Mian County, Hanzhong City, Shaanxi Province, China (33°7′50″ N, 106°48′16″ E). The roots were found at a depth of approximately 20 cm, and the soil was characterized as dark brown loam with a near-neutral pH ranging from 6.55 to 7.05. The soil exhibited a Cd content of 2.15 mg/kg. To isolate rhizosphere bacteria, 5 g of soil was suspended in 10 volumes of sterile phosphate buffer solution and homogenized in a sterile mortar. The resulting suspension was allowed to settle for 5 min, followed by serial dilution and plating onto LB (Luria-Bertan) agar medium (1L) which contained 5 g yeast extract (LP0021, Oxoid, UK), 10 g Tryptone (LP0042, Oxoid, UK), and 5 g NaCl (C111545, Aladdin, Shanghai, China) [60]. Isolated colonies were subsequently subjected to molecular identification via 16S rRNA gene sequencing and phylogenetic analysis. The 16S rRNA gene was amplified using universal bacterial primers 27F (5′-AGAGTTTGATCMTGGCTCAG-3′) and 1492R (5′-GGTTACCTTGTTACGACTT-3′). The PCR amplification protocol included an initial denaturation step at 94 °C for 5 min, followed by 35 cycles of denaturation at 94 °C for 30 s, annealing at 55 °C for 30 s, and extension at 72 °C for 1.5 min, with a final extension at 72 °C for 10 min. Sequencing was performed by Sangon Biotech (Shanghai, China). Sequence homology was analyzed using the BLAST+ 2.11.0 program. Sequence alignment and editing were conducted using MEGA 11.0 software, phylogenetic trees were constructed using the neighbor-joining method [61], and the reliability of the tree topology was evaluated through bootstrap analysis with 1000 replicates.

### 4.2. Screening of Plant Growth-Promoting Traits

The siderophore-producing capacity of bacterial strains was evaluated using the CAS (HB9132, Qingdao, China) agar plate assay [62]. Strains were initially streaked onto LB agar and incubated at 30 °C for 24 h. Single colonies were then transferred onto CAS indicator plates and incubated under identical conditions for 5 days. The formation of an orange or colorless zone surrounding the colonies was recorded as a positive indicator of siderophore production. To assess nitrogen fixation potential, bacterial strains were spread onto nitrogen-free Ashby agar medium and incubated at 30 °C for 7 days [63]. The presence of visible growth, including colorless or pigmented colonies, patches, or biofilm-like structures, was interpreted as evidence of nitrogen-fixing capability. IAA (I101072, Aladdin, Shanghai, China) production was quantified using the Salkowski colorimetric method [64]. Briefly, activated bacterial cultures were inoculated into LB broth supplemented with 500 mg/L tryptophan (L755707, Aladdin, China) and incubated at 30 °C with shaking at 120 rpm for 24 h. Following incubation, 2 mL of each culture was centrifuged at 8000 rpm for 15 min. A 200 μL aliquot of the supernatant was mixed with an equal volume of Salkowski (Dissolve 1 mL of 0.5 mol/L FeCl_3_ (A600454, Song Biotech, Shanghai, China) in 50 mL of 35% perchloric acid (Z1Q0001, Song Biotech, Shanghai, China)) reagent and incubated in the dark for 30 min. The absorbance at 530 nm (OD_530_) and bacterial solution optical density at 600 nm (OD_600_) were measured, with each strain analyzed in triplicate. A reddish color change was considered indicative of IAA production. A 1000 mg/L IAA stock solution was prepared by dissolving IAA in 1 M NaOH (S580606, Aladdin, China) and diluting with sterile water. Serial dilutions were made to obtain concentrations of 0, 2.5, 5, 7.5, 10, 12.5, and 15 mg/L for the standard curve. Each dilution (200 μL) was reacted with Salkowski reagent under dark conditions for 30 min, and the OD_530_ was measured. The resulting calibration curve yielded a regression equation of y = 0.0217x + 0.056 (R^2^ = 0.9903). To determine ACC deaminase, bacterial strains were first cultured in nitrogen-free liquid medium (DF medium) at 30 °C with shaking at 180 rpm for 24 h, followed by transfer to ADF (3 mM ACC (A422646, Aladdin, China) serves as the sole nitrogen source in the ADF medium) solid medium [54,65]. The inoculated plates were incubated at 30 °C for 72 h, with daily observations of growth. Colonies that developed on DF agar were identified as ACC deaminase-positive strains, and concurrent growth on nitrogen-free medium further corroborated the production of this enzyme.

### 4.3. Growth Curve Analysis of Six Bacterial Strains Under Varying Cd Concentrations

Sterilized LB liquid media containing Cd concentrations of 0, 50, 250, and 500 mg/L CdCl_2_ were prepared. Each bacterial strain, previously activated through subculturing, was inoculated into 5 mL of the respective medium in sterile test tubes at a 1% (*v*/*v*) inoculation ratio, The optical density at 600 nm (OD_600_) was monitored every four hours until growth plateaued [66]. The collected growth data were systematically recorded and analyzed to evaluate the growth dynamics of the strains under varying levels of Cd stress.

### 4.4. Co-Culture of LKT25 with S. nigrum

#### 4.4.1. Analysis of Physicochemical Parameters During the Co-Culture of LKT25 and *S. nigrum* Under Varying Soil Cd Concentrations

The co-culture method of LKT25 with *S. nigrum* was adapted from the method proposed by He et al. [39]. The bacterial strain was inoculated onto LB agar plates and incubated at 30 °C for 24 h. A morphologically distinct colony was selected and transferred into 10 mL of LB liquid medium. The culture was then incubated in a shaking incubator at 30 °C and 200 rpm for 24 h. Following cultivation, the bacterial suspension was standardized to a concentration of 10^7^ cfu/mL using sterile normal saline. For inoculation, plant roots were immersed in the prepared bacterial suspension for 5 min, whereas roots in the control group were treated with sterile water under identical conditions.

This study used *S. nigrum* seeds collected from Hanzhong (33°7′50″ N, 106°48′16″ E) and was conducted at the Hanzhong laboratory of Shaanxi University of Technology. The soil, classified as brown-yellow soil (Xanthic Ali-Udic Cambisols of silt loam textural class according to WRB, 2015 classification), was sampled from the top layer (0–20 cm) on campus. It had a pH of 6.5, organic matter content of 26.8 g/kg, total nitrogen of 1.71 g/kg, available potassium of 17.3 mg/kg, available phosphorus of 16.8 mg/kg, and Cd content of 0.08 mg/kg. According to GB 15618–2018, the soil Cd level is Class I, representing a natural background concentration [67].

The pot experiments were conducted following the method described by Dai et al. [68]. Soil was spiked with an analytical-grade reagent solution of CdCl_2_·2.5H_2_O (CdCl_2_·2.5H_2_O, Aladdin, China) and equilibrated for two months prior to the initiation of the experiment. Treatments included one control group (CK, 0 mg/kg) and three Cd levels (5, 25, and 50 mg/kg), each with three replicates.

Each pot (25 cm × 15 cm) was filled with 2.5 kg of equilibrated soil. *S. nigrum* seeds were surface-sterilized with 0.1% HgCl_2_ (Shanghai, China) for 10 min before sowing. After germination, seedlings were grown under natural light. Four uniform seedlings (~5 cm tall) were transplanted into each pot and cultivated in a greenhouse environment characterized by a 16/8 h photoperiod (day/night), a temperature regime of 25/20 °C (day/night). The plants were watered regularly with deionized water to maintain soil moisture at approximately 75% of field capacity.

During the 50-day cultivation period, the treatment group received root irrigation with 10 mL of bacterial suspension at 8-day intervals, for a total of three applications, while the control group received sterile water. At maturity, all plants were carefully uprooted and growth parameters were measured.

For biomass (dry weight) determination, plant samples were collected, thoroughly rinsed with deionized water to remove surface contaminants, blotted dry with absorbent paper, and partitioned into roots, stems, and leaves. The samples were first subjected to a rapid inactivation step at 105 °C for 5 min in a drying oven to prevent enzymatic degradation, followed by drying at 80 °C until a constant weight was achieved [69].

The Cd content in *S. nigrum* tissues was quantitatively analyzed using atomic absorption spectrophotometry (Hitachi 180-80, Tokyo, Japan) to determine the distribution of Cd concentrations in both root and shoot biomass. For quality assurance and quality control (QA/QC), measurement accuracy was validated using certified reference material GBW07405 (GSS-5), following the protocol described by Dai et al. [68]. The calculation formulas for Cd distribution across different organs of *S. nigrum* are as follows [68]: Bioaccumulation factor = (Cd concentration in shoot)/(Cd concentration in the soil). Translocation factor = (Cd concentration in shoot)/(Cd concentration in root). Removal efficiency = (Cd concentration in the soil after *S. nigrum* planting/Cd concentration in the soil before planting) × 100%.

Total chlorophyll content in *S. nigrum* leaves was quantified using the spectrophotometric method of Dai et al. [70]. The net photosynthetic rate of fully expanded leaves was measured in controlled environmental conditions using a CI-340 portable photosynthesis system (CID Inc., New York, NY, USA) [70].

#### 4.4.2. Determination of Antioxidant Enzyme Activities in Different Tissues of *S. nigrum*

Fresh tissue samples of Solanum nigrum roots and stems (1 g) were equally divided into two portions. These were then homogenized on ice using a 50 mM phosphate buffer (pH 7.8) containing 5.0 mM ascorbic acid (employed for the determination of APX enzyme activity) and a 50 mM phosphate buffer (pH 7.8) without ascorbic acid, respectively [70]. The resulting homogenate was transferred to a volumetric flask and diluted to a final volume of 10 mL. After centrifugation at 12,000× *g* for 20 min at 4 °C, the supernatant was collected and used as the enzyme extract. The activities of SOD, POD, CAT and APX in both shoots and roots of *S. nigrum* were quantified using a commercial biochemical assay kit (Song Biotech, Shanghai, China), in accordance with the manufacturer’s instructions. The malondialdehyde (MDA) content in these tissues was determined following the methodology of Dai et al. [70], with all measurements followed the manufacturer’s instructions in the kit manual to ensure data consistency and reliability.

### 4.5. Statistical Analysis

All data were presented as mean ± standard deviation (SD). Statistical analysis was performed using one-way analysis of variance (ANOVA) with SPSS 13.0 software. Significant differences among treatment groups were defined as * *p* < 0.05 and ** *p* < 0.01. Plant growth and physiological parameters were evaluated in biological triplicate, whereas microbial characteristics were analyzed using five biological replicates. Data visualization and figure generation were conducted using Origin 2024 software.

## 5. Conclusions

This study reports the isolation and identification of a novel *Bacillus thuringiensis* strain, LKT25, from *S. nigrum* rhizosphere soil. The strain exhibits key plant growth-promoting traits, including IAA production, nitrogen fixation, ACC deaminase, and siderophore synthesis. When co-cultured with *S. nigrum*, LKT25 significantly improves plant growth, photosynthesis, and Cd removal across tested concentrations. The strain shows different responses to Cd stress: it promotes Cd translocation to shoots at low concentrations and enhances root retention at high concentrations. These results suggest that LKT25 is a promising microbial agent for bioremediation, especially in high-Cd environments.

## Figures and Tables

**Figure 1 plants-14-02918-f001:**
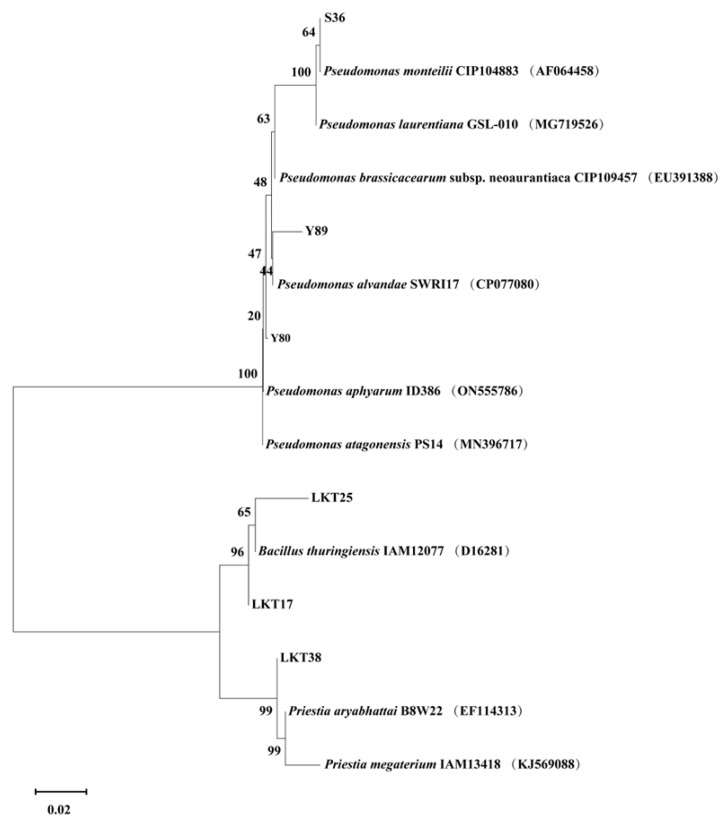
Phylogenetic analysis of the six bacterial strains.

**Figure 2 plants-14-02918-f002:**
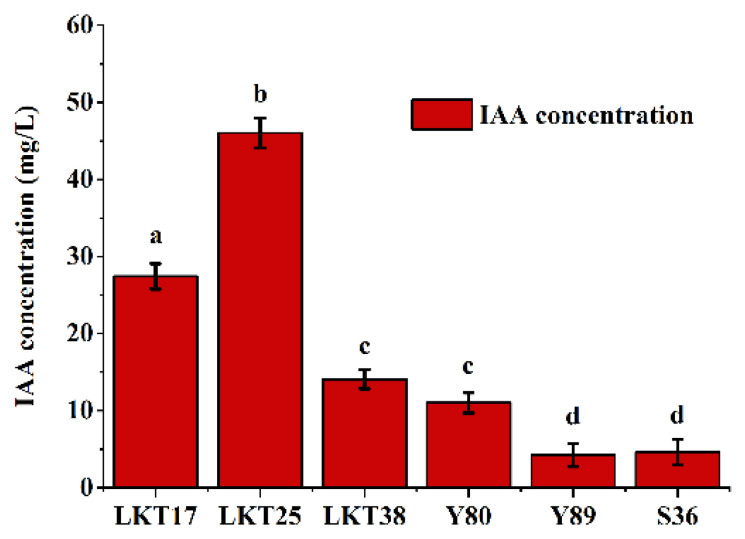
Quantitative determination of IAA production by the six bacterial strains. Lowercase letters denote statistically significant differences in IAA production among various bacterial strains (*p* < 0.05).

**Figure 3 plants-14-02918-f003:**
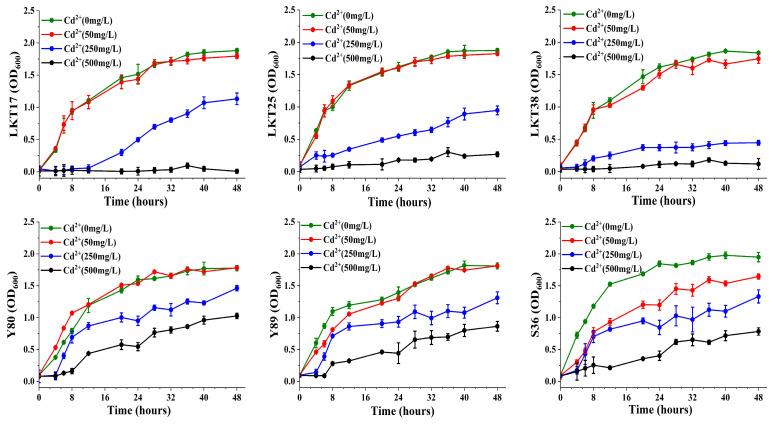
Assessment of growth performance of the six bacterial strains under different Cd stress conditions.

**Figure 4 plants-14-02918-f004:**
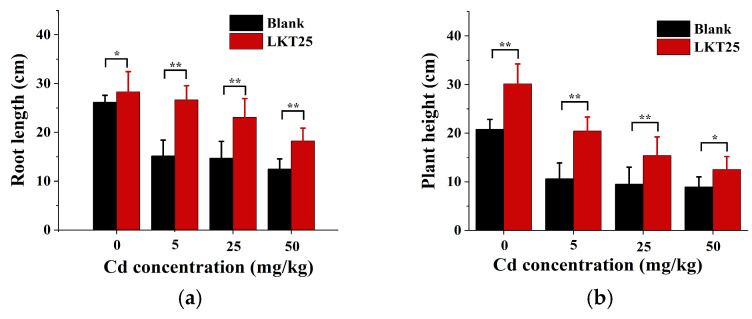
The impact of LKT25 on key growth and development parameters of *S. nigrum* under varying soil Cd concentrations. (**a**) The impact of LKT25 on root length of *S. nigrum* under varying soil Cd concentrations. (**b**) The impact of LKT25 on plant height of *S. nigrum* under varying soil Cd concentrations. (**c**) The impact of LKT25 on dry weight in shoots of *S. nigrum* under varying soil Cd concentrations. (**d**) The impact of LKT25 on dry weight in roots of *S. nigrum* under varying soil Cd concentrations. *, ** indicate statistically significant differences at *p* < 0.05, *p* < 0.01, respectively.

**Figure 5 plants-14-02918-f005:**
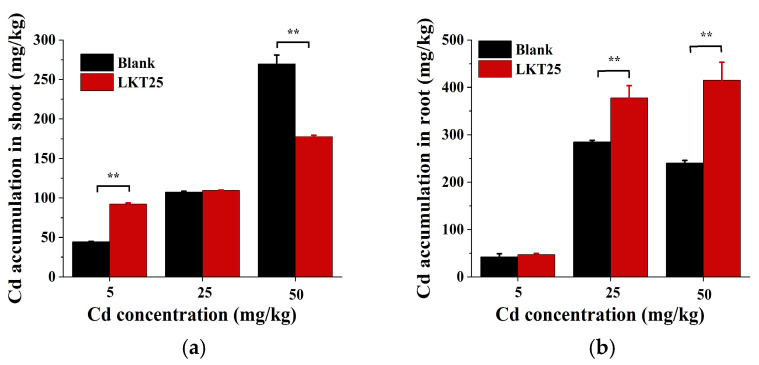
The effect of LKT25 on Cd accumulation of the *S. nigrum*. (**a**) The effect of LKT25 on Cd accumulation in the shoots of *S. nigrum*. (**b**) The effect of LKT25 on Cd accumulation in the roots of *S. nigrum*. ** indicates statistically significant differences at *p* < 0.01.

**Figure 6 plants-14-02918-f006:**
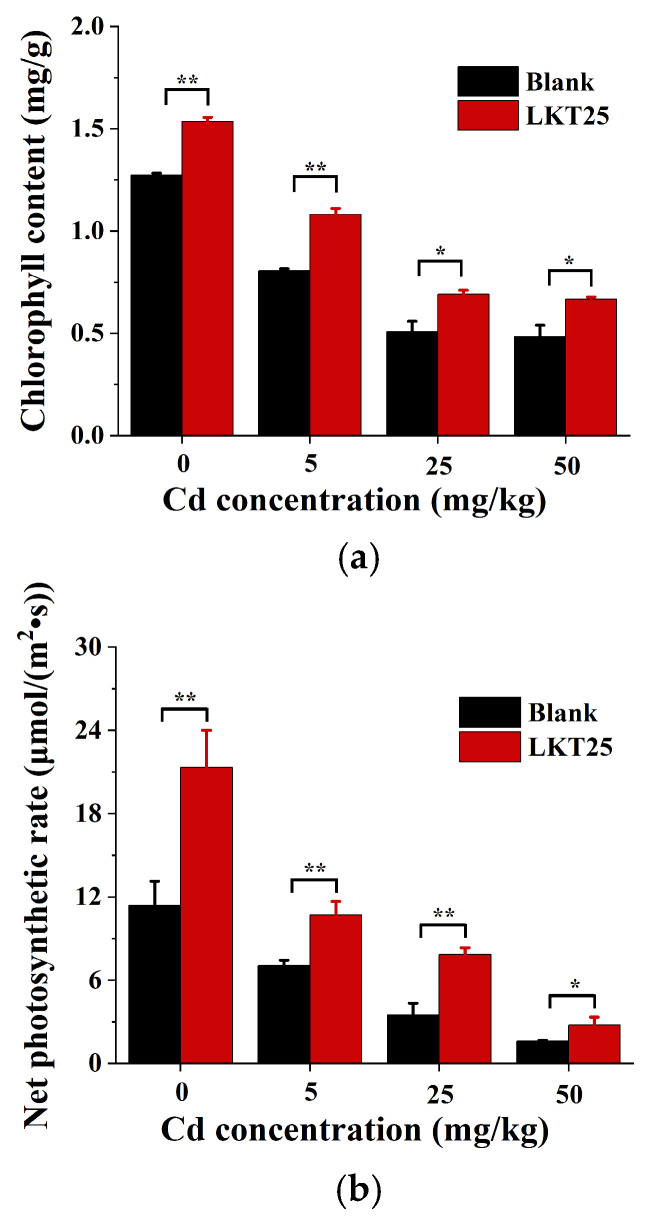
The effect of LKT25 on photosynthetic parameters of the *S. nigrum*. (**a**) The effect of LKT25 on chlorophyll of the *S. nigrum*. (**b**) The effect of LKT25 on photosynthetic rate of the *S. nigrum*. *, ** indicate statistically significant differences at *p* < 0.05, *p* < 0.01, respectively.

**Figure 7 plants-14-02918-f007:**
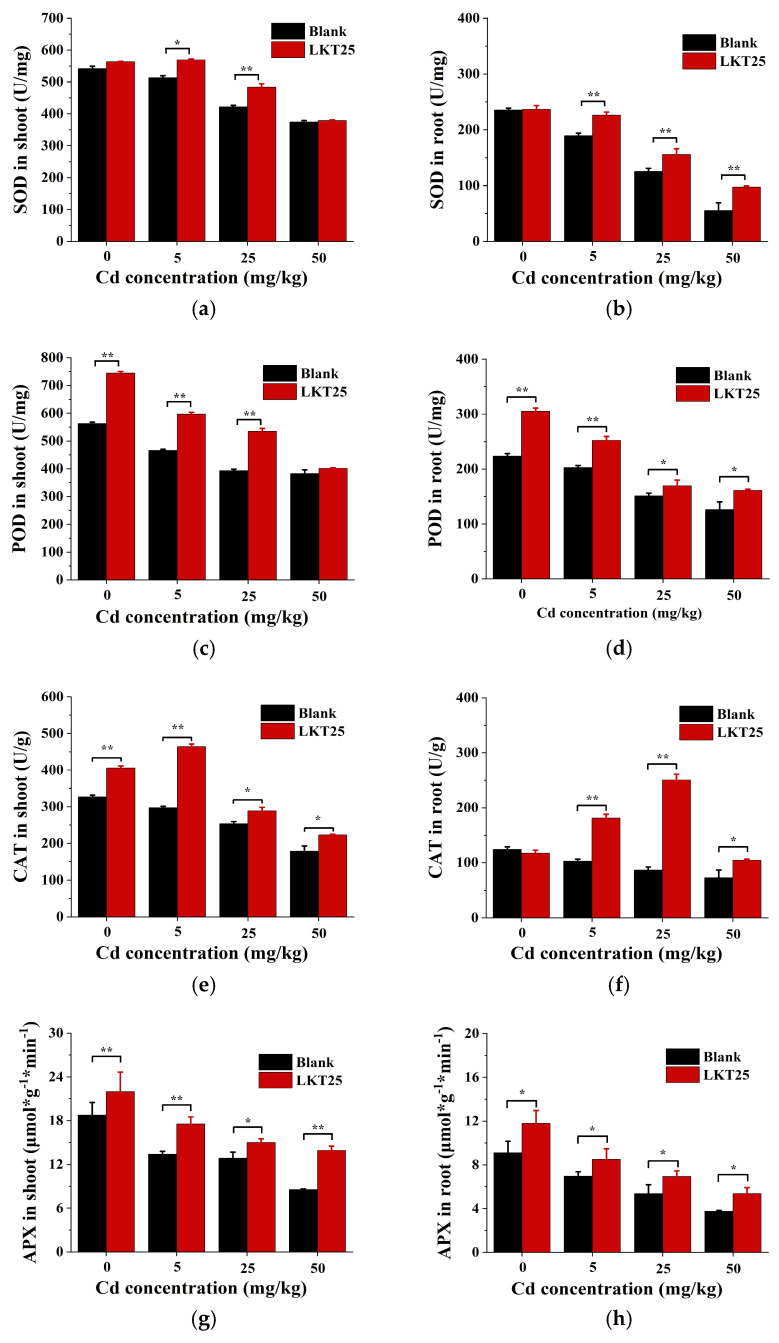
The effect of LKT25 on the antioxidant enzyme activity of *S. nigrum*. (**a**) The effect of LKT25 on the SOD activity in the shoots of *S. nigrum*. (**b**) The effect of LKT25 on the SOD activity in the roots of *S. nigrum*. (**c**) The effect of LKT25 on the POD activity in the shoots of *S. nigrum*. (**d**) The effect of LKT25 on the POD activity in the roots of *S. nigrum*. (**e**) The effect of LKT25 on the CAT activity in the shoots of *S. nigrum*. (**f**) The effect of LKT25 on the CAT activity in the roots of *S. nigrum*. (**g**) The effect of LKT25 on the APX activity in the shoots of *S. nigrum*. (**h**) The effect of LKT25 on the APX activity in the roots of *S. nigrum*. *, ** indicate statistically significant differences at *p* < 0.05, *p* < 0.01, respectively.

**Figure 8 plants-14-02918-f008:**
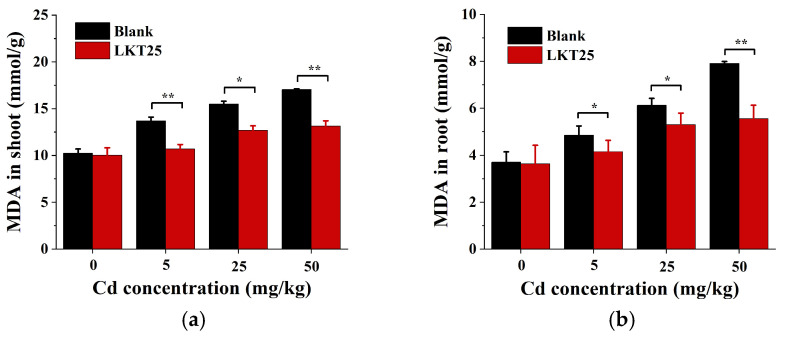
The effects of LKT25 on MDA levels in the plant *S. nigrum*. (**a**) The effects of LKT25 on MDA in shoot of the plant *S. nigrum*. (**b**) The effects of LKT25 on MDA in root of the plant *S. nigrum*. *, ** indicate statistically significant differences at *p* < 0.05, *p* < 0.01, respectively.

**Table 1 plants-14-02918-t001:** The plant growth promotion traits of bacterial isolates.

Strain ID	ACC Deaminase	Siderophore Production	Nitrogenase
LKT25	+	+	+
LKT17	−	+	+
LKT38	−	+	+
Y80	−	+	+
Y89	−	+	+
S36	−	+	+

**Table 2 plants-14-02918-t002:** Effects of LKT25 application on Cd accumulation parameters in *S. nigrum*.

Treatment		Enrichment Coefficient		Transfer Coefficient	Cd Removal Rate (%)
		Shoot	Root	Shoot	
5 mg/kg (Cd)	Blank	8.87 ± 0.43	8.48 ± 0.31	1.05 ± 0.14	31.12
	LKT25	18.40 ± 1.4 **	9.44 ± 0.49	1.95 ± 0.26 *	45.13 *
25 mg/kg (Cd)	Blank	4.29 ± 0.15	11.40 ± 0.58	0.38 ± 0.03	15.32
	LKT25	4.38 ± 0.06	15.09 ± 0.14 *	0.31 ± 0.02	22.95 *
50 mg/kg (Cd)	Blank	5.39 ± 0.05	4.81 ± 0.23	1.12 ± 0.03	21.75
	LKT25	3.55 ± 0.05 *	8.30 ± 0.35 **	0.42 ± 0.14 **	27.86

*, ** indicate statistically significant differences at *p* < 0.05, *p* < 0.01, respectively.

## Data Availability

Data are contained within the article.

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
