# Peer review of "The Effect of Plant Growth Promoting Rhizobacteria Bacillus thuringiensis LKT25 on Cadmium Accumulation and Physiological Responses in Solanum nigrum L."

_plants, 2025, doi:10.3390/plants14182918_

Round 1
Reviewer 1 Report
Comments and Suggestions for Authors
The manuscript presents the results of Cd-phytoremediation tests conducted on Solanum nigrum treated with the B. thuringensis PGPR isolate. The experimental design was appropriate and all control samples were included. The results are clearly described and commented. I noted references to supplementary materials that were not provided in the version of the manuscript submitted for revision. Other detailed comments are as follows:
Line 19: mg/L or mg/kg?
Lines 27-81: The introduction is short, but covers the key aspects of the studied phenomenon. In my opinion, it should be supplemented with information on typical Cd contamination levels; without such context it is difficult to accurately interpret the obtained results.
Lines 91-93 and later 256-257: Figures were not attached to the manuscript
Line 100 (and later): Figures - particularly Figure 1 - are too small, making the legends and axis labels nearly illegible without significant magnification. Although the overall quality is acceptable, they should be enlarged in the final version of manuscript or (for Fig. 1) possibly divided into three separate figures.
Line 123: it is “an d”, should be “and”
Line 128 (and elsewhere in other figure descriptions): “*, ** indicate significant correlations at P<0.05, P<0.01 and P<0.001, respectively” – if one asterisk refers to p<0.05, and two asterisks to P<0.01, what symbol denotes P<0.001?
Line 142: it is “a cumulation”, should be “accumulation”
Line 170: it is “Table 1 Effects…”, should be “Table 1. Effects…”
Lines 179-180: There appears to be no significant inhibition decrease between Cd concentrations of 25 and 50 mg (as reported in Fig. 4)
Line 200 (and subsequently for similar cases): please expand all abbreviations upon first use
Lines 240-280: The discussion is brief and largely reads as a summary of the results. Only four original papers were cited. This section requires substantial revision, including comparison of the obtained results with those reported in similar studies. Without such comparison, it is difficult to determine whether the results are high or low in the context of common soil contamination levels, Cd removal rates and other coefficients reported by other authors, possibly for plants other than S. nigrum.
Lines 317: Why was OD600 measured for IAA production quantification?
Line 345: I question whether 107 cfu/mL of microorganisms is sufficient to significantly affect the plant rhizosphere. What was the indigenous microbiota density in the soil used for the experiment? Was it assessed whether the introduced strain integrated into the native microbiota and remained detectable in the soil after the experiment?
Line 349: How many applications of the bacterial suspension were performed?
Line 351-352: How long did the experiment last? What were the cultivation conditions - greenhouse or field?
Line 364: please correct citation number (marked as 41) and provide the corresponding reference
Lines 373-379: The CAT methodology is not described.
Line 379: Please clarify what is meant by “standardized experimental conditions”
Line 383: The figure descriptions contain p-value notations that differ from those described in the text - please unify.
Author Response
We sincerely appreciate the considerable time and effort you have dedicated to reviewing our manuscript, as well as your comprehensive, insightful, and constructive feedback. Your comments have significantly enhanced the depth of our research and the overall quality of the manuscript. We have carefully addressed each of your suggestions through thorough revisions and supplementary content. Below, we provide point-by-point Response to your comments, detailing all modifications implemented accordingly.
Comment 1: Line 19: mg/L or mg/kg?
Response 1: We sincerely appreciate your identification of this critical unit discrepancy. Please accept our genuine apologies for this oversight. In the soil cultivation experiment, the unit for Cd concentration should indeed be expressed as mg/kg. We have thoroughly verified and corrected all concentration units to mg/kg accordingly. Please refer to Line 18 of the revised manuscript for the specific amendment.
Comment 2: Lines 27-81: The introduction is short, but covers the key aspects of the studied phenomenon. In my opinion, it should be supplemented with information on typical Cd contamination levels; without such context it is difficult to accurately interpret the obtained results.
Response 2: This is an extremely valuable suggestion. We fully concur that presenting representative Cd pollution levels as contextual background is essential for accurately interpreting the significance of our experimental findings (5-50 mg/kg). In the Introduction section (revised manuscript, lines 33-42), we have incorporated typical Cd concentration ranges observed in soils from various regions, including mining areas and farmlands, and supported these data with references to multiple authoritative studies. This contextual information provides a scientifically sound basis for the selection of our experimental concentration levels.
The specific content is as follows: “At present, Cd pollution in China represents the most severe form of heavy metal contamination, with 7.0% of monitored sites exceeding the threshold outlined in the Soil Environmental Quality Standard (GB15618-1995) issued by the Ministry of Ecology and Environment of the People's Republic of China under pH conditions of 6.5-7.5 (0.6 mg/kg) [9]. Notably, Cd concentrations exceeding 3 mg/kg surpass the risk control threshold, significantly elevating the likelihood of contamination in agricultural products (GB15618-1995). Particularly alarming is the situation in mining areas, where Cd levels have been recorded at over 80 mg/kg [10]. The remediation of Cd in agricultural soils constitutes a critical environmental challenge that demands urgent and effective intervention strategies [10]”.
Comment 3: Lines 91-93 and later 256-257: Figures were not attached to the manuscript
Response 3: We sincerely regret this oversight. In the revised manuscript, A new Table 1 has been inserted at line 128 of the main text, and supplementary materials have been provided for your reference and evaluation.
Comment 4: Line 100 (and later): Figures - particularly Figure 1 - are too small, making the legends and axis labels nearly illegible without significant magnification. Although the overall quality is acceptable, they should be enlarged in the final version of manuscript or (for Fig. 1) possibly divided into three separate figures.
Response 4: We appreciate your valuable feedback. Figure 1 has been reprocessed and divided into three distinct figures (Figures 1, 2, and 3), each featuring an appropriately large size and clearly legible annotations. The corresponding locations of these figures in the revised manuscript are indicated at lines 116, 125, and 135, respectively.
Comment 5: Line 123: it is “an d”, should be “and”
Response5: We sincerely appreciate your meticulous proofreading. The spelling error has been rectified and the correction can be found at line 154 of the revised manuscript.
Comment 6: Line 128 (and elsewhere in other figure descriptions): “*, ** indicate significant correlations at P<0.05, P<0.01 and P<0.001, respectively” – if one asterisk refers to p<0.05, and two asterisks to P<0.01, what symbol denotes P<0.001?
Response 6: We appreciate your valuable comment regarding the unclear description. The notation for statistical significance has been standardized throughout the manuscript. In all figure captions and textual explanations, the following statement has been explicitly included: "*, ** denote statistically significant differences at P < 0.05 and P < 0.01, respectively."
Comment 7: Line 142: it is “a cumulation”, should be “accumulation”
Response 7: The spelling error has been rectified and the correction is now reflected at line 173 of the revised manuscript.
Comment 8: Line 170: it is “Table 1 Effects…”, should be “Table 1. Effects…”
Response 8: In accordance with your suggestion, a period has been added to the table title. Due to the addition of a new table to the manuscript, the original table numbering has been updated, and the relevant table is now designated as Table 2. The revised manuscript can be found on line 200.
Comment 9: Lines 179-180: There appears to be no significant inhibition decrease between Cd concentrations of 25 and 50 mg (as reported in Fig. 4)
Response 9: We sincerely appreciate your keen observation. We have conducted a reanalysis of the data and reviewed Figure 4 (as the original Figure 1 has been divided into three separate figures, the corresponding Figure 4 in the revised manuscript has been renumbered as Figure 6, and all figure references throughout the text have been updated accordingly). Your comment is highly accurate: the changes in inhibition rates of certain parameters between the two highest concentration levels are not statistically significant. We have revised the overly definitive statements in the original text and now more precisely state: " Exposure to Cd significantly suppresses chlorophyll synthesis (Fig. 4)." Thank you for your thorough and thoughtful review of our work, which has contributed to improving the precision of our results' interpretation.
Comment 10: Line 200 (and subsequently for similar cases): please expand all abbreviations upon first use
Response 10: We have conducted a comprehensive review of the entire manuscript. The full terms for all abbreviations, including IAA, ACC, SOD, CAT, POD, and APX, have been clearly defined upon their first appearance in the text. For your reference, the corresponding lines are 70, 71, 75, and 81 in the revised manuscript.
Comment 11: Lines 240-280: The discussion is brief and largely reads as a summary of the results. Only four original papers were cited. This section requires substantial revision, including comparison of the obtained results with those reported in similar studies. Without such comparison, it is difficult to determine whether the results are high or low in the context of common soil contamination levels, Cd removal rates and other coefficients reported by other authors, possibly for plants other than S. nigrum.
Response 11: This section represents the most intensive effort in the current revision. We fully concur with the assessment that the previous discussion section was overly concise. Accordingly, we have comprehensively rewritten and substantially expanded this section. The key revisions are outlined as follows:
Enhanced Comparative Analysis: Our research findings-such as increases in biomass, Cd remediation efficiency, transport factor, and variations in antioxidant enzyme activity have been thoroughly compared with both domestic and international studies on S. nigrum, as well as with research on other commonly studied heavy metal hyperaccumulator plants.
Mechanistic Interpretation: Drawing upon relevant literature, we have elaborated on the potential mechanisms by which PGPR inoculation may mitigate Cd toxicity and enhance plant growth. These mechanisms include the secretion of IAA, nitrogen fixation, production of ACC deaminase and siderophores, activation of the antioxidant enzyme system, and the expression of plant genes associated with heavy metal accumulation and transport moving beyond a mere descriptive account of observed phenomena.
Assessment of Remediation Performance: In conjunction with the background data on typical cadmium contamination levels, we have objectively evaluated the remediation efficiency and application potential of strain LKT25 at Cd concentrations of 5, 25, and 50 mg/kg.
Expanded Literature Support: The number of references cited in the discussion section has been significantly increased from 4 to 32, incorporating key and recent studies in the field to substantiate our arguments and comparative analyses. The revised discussion section is presented as follows (lines 267-352 in the revised manuscript):
“3.1. LKT25 enhances the growth and photosynthetic efficiency of S. nigrum
PGPR significantly enhance plant growth performance by facilitating nutrient uptake and suppressing the proliferation of pathogenic bacteria and pests [29–30]. In this study, bacterial strain LKT25 was identified as Bacillus thuringiensis through 16S rRNA gene sequence analysis. The sequence exhibited a similarity of 98.8% to that of the reference strain Bacillus thuringiensis ATCC 10792 (Figure 1), indicating that it represents a novel strain not previously reported in the literature. Excessive accumulation of heavy metals can exert detrimental effects on plant physiology, either directly or indirectly, leading to growth inhibition [31-32]. Bacillus has great potential in the production of auxins, lytic enzymes, siderophores and nitrogen fixation and these growth-promoting factors can enhance plant growth and increase plant's stress resistance through direct or indirect means [33-35]. Among these factors, IAA, at optimal concentrations, promotes the development of xylem and root systems, stimulates the synthesis of photosynthetic pigments, and enhances photosynthetic efficiency [36-37]. Additionally, the synthesis of ACC deaminase synergizes with IAA to facilitate root elongation under short-term conditions [33]. Nitrogen-fixing PGPR capable of biological nitrogen fixation can supply essential nitrogen nutrients to plants, thereby promoting plant growth [33]. In this study, strain LKT25 not only exhibits a high capacity for IAA production (as shown in Figure 2), but also possesses the abilities to synthesize ACC deaminase (Table 1 & Supplementary Material Figure A3) and to perform nitrogen fixation (Table 1 & Supplementary Material Figure A1). The IAA yield achieved by this strain was 46.03 mg/L, which was significantly greater than that of other Cd-tolerant strains analyzed in this research (as shown in Figure 2), and also exceeded the previously reported IAA production range for other strains within the same genus (5.7–22.9 mg/L) [38–39]. Furthermore, LKT25 significantly stimulated root development and biomass in S. nigrum (Figure 4a, c, d), with the most pronounced effects observed under a soil Cd concentration of 5 mg/kg. Under this condition, root length increased by 1.75-fold and biomass by 2.69-fold compared to the control group, while the shoot biomass reached 6.6 times that of the control group, surpassing the levels reported for other species within the Bacillus genus [40] and a variety of other bacteria [27, 41-42]. Additionally, the application of LKT25 significantly increased the total chlorophyll content and improved photosynthetic efficiency in S. nigrum (Figure 6a & b). These findings align with previous studies [3, 3-40], showing that LKT25 significantly reduces the inhibitory effects of Cd stress on photosynthesis and promotes plant growth.
3.2. LKT25 enhances the antioxidant system of S. nigrum and reduces Cd-induced oxidative stress
Plant defense against heavy metal stress depends on their ability to maintain redox homeostasis in the antioxidant system [43-44]. Heavy metal exposure triggers ROS overproduction, leading to lipid peroxidation and the formation of cytotoxic MDA [45]. Our results show that LKT25 significantly reduces MDA levels in S. nigrum under all tested Cd concentrations (Figures 8a and 8b), indicating that co-cultivation with LKT25 effectively alleviates oxidative stress.
Plant-microbe co-cultivation mainly enhances antioxidant capacity by regulating antioxidant enzymes and non-enzymatic substances [46-47]. Enzymes like SOD, POD, CAT, and APX help maintain protein stability, preserve membrane integrity, and reduce lipid peroxidation [48]. Our results show that LKT25 significantly boosts antioxidant enzyme activity (SOD, POD, CAT, APX) in S. nigrum roots at all tested Cd levels (Figure 7 b, d, f, h), reducing oxidative stress. At 0, 5, 25mg/kg tested Cd concentrations, LKT25 also enhances these enzymes in the shoots (Figure 7 a, c, e, g). However, at 50 mg/kg, no significant difference in stem SOD and POD activity is found between treated and control plants (Figure 7 b, d). This may be due to lower Cd accumulation in the shoots of LKT25-treated plants (177.6 mg/kg vs. 269.8 mg/kg in controls, Figure 5 a, b), suggesting that LKT25 mainly acts in the roots and limits Cd movement upward, resulting in no strong stress response in the stems.
3.3. LKT25 regulates the accumulation and transport of Cd in S. nigrum
Hyperaccumulator plants are key to bioremediation because they can greatly reduce heavy metal levels in soil through excessive accumulation [16]. However, high pollution levels often limit their effectiveness [49]. PGPR have been shown to promote plant growth and enhance Cd absorption [27,50]. Microbial-plant co-cultivation systems use different strategies to reduce Cd stress depending on pollution levels [51]. Under low contamination, plant growth-promoting bacteria help plants absorb and transport heavy metals to aboveground parts through processes like IAA secretion, nitrogen fixation, ACC deaminase production, siderophore synthesis [52-53], and organic acid release [33-37]. In high contamination, Cd mainly accumulates in roots (root > stem > leaf), with limited movement to shoots, reducing toxicity [54-55]. In this case, plant-microbe interactions regulate the production of compounds like ACC deaminase, antioxidant enzymes, and non-enzymatic antioxidants to reduce oxidative damage [48,55-56]. This study shows that LKT25 significantly improves the Cd removal ability of S. nigrum across pollution levels. At 5 mg/kg soil Cd, LKT25 achieved a 45.13% removal rate (Figure 5a and 5b), matching current rhizobacteria-assisted phytoremediation methods [39–40, 42, 57-58]. It also increased shoot Cd by 2.07 times compared to the control, with a transfer coefficient over 1-higher than other Bacillus strains [40, 55-56]. At this stage, a favorable interaction between LKT25 and S. nigrum facilitated the absorption and upward translocation of Cd, indicating its significant potential for practical application. At a Cd concentration of 25 mg/kg, S. nigrum restricts Cd translocation from roots to shoots via intrinsic regulatory mechanisms, resulting in a transport coefficient below 1. This condition showed the lowest Cd removal efficiency, lower than at 5 mg/kg and 50 mg/kg. Applying LKT25 significantly increased root Cd accumulation and elevated oxidative stress levels, effectively protecting aboveground tissues and improving overall Cd removal. In contrast, under 50 mg/kg Cd stress, S. nigrum absorbed more Cd than at 25 mg/kg, suggesting strong activation of defense and tolerance mechanisms. This stress may up-regulate genes involved in heavy metal absorption, translocation, and compartmentalization [59-60], promoting Cd movement to shoots, with a transport coefficient of 1.12-higher than the 1.05 observed at 5 mg/kg. Under this condition, LKT25 further increased root Cd enrichment and reduced shoot translocation (coefficient < 1), alleviating plant stress. These results show that LKT25 not only protects plants from Cd-induced damage but also has strong synergistic potential in high-Cd environments, with possible effectiveness at concentrations above 50 mg/kg.”
Comment 12: Lines 317: Why was OD600 measured for IAA production quantification?
Response 12: Your inquiry highlights an important point. The measurement of OD600 is employed to standardize bacterial growth across experimental conditions. A corresponding explanation has been added to the revised manuscript (line 389).
Comment 13: Line 345: I question whether 107 cfu/mL of microorganisms is sufficient to significantly affect the plant rhizosphere. What was the indigenous microbiota density in the soil used for the experiment? Was it assessed whether the introduced strain integrated into the native microbiota and remained detectable in the soil after the experiment?
Response 13: Response: We sincerely appreciate the reviewer's insightful and significant inquiry, which addresses a critical aspect of our research.
First and foremost, we acknowledge that in the current study, we did not specifically monitor or quantify the colonization dynamics of the inoculated strain Bacillus thuringiensis LKT25 in the rhizosphere. This constitutes a limitation in our experimental design, and we are grateful to the reviewer for highlighting this issue.
Nonetheless, we maintain that the inoculation was effective, based on the following considerations:
- Rationale for the Inoculation Concentration:
The inoculation concentration selected in our study (10⁷ CFU/mL) is a commonly used and well-documented dosage in PGPR research, with its efficacy supported by extensive scientific literature. [reference: Yang, M.; Fu, Y.; Hu, S.; Leng, F.; Zhuang, Y.; Sun, W.; Wang, Y. Potential function of plant-growth-promoting endophytic Serratia fonticola CPSE11 from Codonopsis pilosula in phytoremediation of Cadmium ion (Cd2+). J. Environ. Manage. 2025, 380, 124994.]. This concentration is intended to ensure an adequate initial bacterial load to effectively compete for ecological niches within the rhizosphere.
- Indigenous Microbial Density:
The experimental soil was non-sterilized, with a background density of culturable indigenous bacteria estimated at approximately 10⁶ CFU/g using the dilution plate method. Therefore, the applied inoculum level was sufficiently high to induce measurable changes in rhizosphere microbial composition within a short timeframe.
- Indirect Evidence of Colonization Success:
Although direct colonization data were not obtained, the consistent and statistically significant physiological Response observed including enhanced plant biomass, altered antioxidant enzyme activities, and increased Cd accumulation serve as compelling indirect evidence that the inoculated strain successfully colonized and exerted functional effects in the rhizosphere. Such pronounced phenotypic changes would be unlikely if the strain had failed to establish itself in the root environment.
We fully concur with the reviewer that direct monitoring of the inoculated strain’s colonization behavior is essential for elucidating its underlying mechanisms of action. This will be a central focus of our future work, where we plan to employ molecular tools such as GFP labeling, qPCR with strain-specific primers, or antibiotic resistance markers to precisely track the spatial and temporal dynamics of strain LKT25 within the soil-plant system.
Comment 14: Line 349: How many applications of the bacterial suspension were performed?
Response 14: The bacterial suspension was administered via root drenching and inoculation three times following transplantation. This information has been explicitly clarified in the Materials and Methods section (lines 441-444 in the revised manuscript).
Comment 15: Line 351-352: How long did the experiment last? What were the cultivation conditions - greenhouse or field?
Response 15: We sincerely apologize for the omission of these critical details and have now incorporated them into the manuscript.
The experimental duration was set to 50 days post-transplantation (line 441 in the revised manuscript).
Regarding cultivation conditions, plants were grown in a controlled greenhouse environment characterized by a 16/8-hour photoperiod (day/night), a temperature regime of 25/20 °C (day/night), and a relative humidity level of 75% (lines 438-440 in the revised manuscript).
Comment 16: Line 364: please correct citation number (marked as 41) and provide the corresponding reference
Response 16: The reference list at the end of the manuscript has been reviewed and revised to ensure accurate alignment between citation number 41 and its corresponding in-text reference (line 454 in the revised manuscript).
Comment 17: Lines 373-379: The CAT methodology is not described.
Response 17: Thank you for your valuable feedback. The measurement of CAT activity was performed using the identical commercial kit as that used for POD and SOD (Song Biotech, Shanghai, China). The assay methodology has been detailed in conjunction with the procedures for POD and SOD. The specific description is presented below: “The activities of SOD, POD, CAT and APX in both shoots and roots of S. nigrum were quantified using a commercial biochemical assay kit (Song Biotech, Shanghai, China), in accordance with the manufacturer’s instructions.”
Comment 18: Line 379: Please clarify what is meant by “standardized experimental conditions”
Response 18: We have clarified this ambiguous statement by revising it to: “all measurements followed the manufacturer's instructions in the kit manual to ensure data consistency and reliability.” (Lines 473-474 in the revised manuscript).
Comment 19: Line 383: The figure descriptions contain p-value notations that differ from those described in the text - please unify.
Response 19: We have conducted a comprehensive review and standardized all p-value significance notations across both the figure captions and main text to ensure full consistency (adopting the standardized format: *P < 0.05, **P < 0.01). (The revised manuscript can be found on line 478).

Reviewer 2 Report
Comments and Suggestions for Authors
The main problems with this manuscript are related to the methodology section and discussion. The study is not replicable because essential information about the conditions and actions taken is missing. In addition, there are fundamental errors in the description of the methodology, which suggest that the authors did not perform some of the described actions. The discussion does not explain the main result highlighted – why does bacterial treatment alter Cd uptake?
Title
Only the effect of the bacteria on Cd accumulation is emphasized, which is inherently contradictory, so the title does not match the content. In fact, the only clear result is the improvement in growth and changes in the antioxidant enzymatic system under the influence of the bacteria.
Abstract
Do not introduce abbreviations for terms not used more than three times within the abstract. ACC deaminase activity was not measured.
Introduction
Mostly too general. Previous similar studies not analyzed in detail, as He et al. (2020) Ecotoxicol. Environ. Safety and Chi et al. (2023) J. Hazard. Mater. Therefore, it is not becoming clear why another study was necessary. The aim is not clearly formulated.
Several sentences describing particular facts have no references (lines 30–31; lines 56–57; lines 72–74)
Results
Do not include facts from literature with references (lines 174–176; line 228).
In legends to figures 2 to 6, instead of "statistically significant differences", "significant correlations" is indicated.
Discussion
Too short and inconsistent with the results obtained.
The discussion places inappropriate emphasis, omitting conflicting information about the different effects of bacterial treatments on Cd accumulation. There is no explanation why at the lowest Cd concentration the bacterium promoted its accumulation in the shoot, at the medium concentration it had no effect, and at the highest concentration it reduced it. This is in no way related to the observed changes in enzyme activity, which were mostly positive.
Contradictions in results from the similar studies need to be discussed.
Materials and methods
The weakest section, which raises doubts about the scientific integrity of the study.
In 4.2, it is claimed that IAA was dissolved in distilled water, but this compound is insoluble in water.
Lines 324–329 as if describe measurement of "ACC deaminase" activity. No activity was detected here. In fact, the provided reference is to composition of the original medium used by Dworkin and Foster. For assessment of bacteria, capable for using ACC as a nitrogen source through deaminase activity, ACC needs to be added to the medium. To measure deaminase activity itself, different methods can be used, but the simplest way is to record ACC breakdown rate using nonspecific ninhydrin assay.
In 4.4.1, no information is provided to understand how plant material was obtained, propagated, what were edaphic and other conditions (temperature, photon flux density of photosynthetically active radiation, photoperiod, relative humidity, fertilization, watering etc.) for plant cultivation, how long plants were cultivated etc.
In 4.4.2 it needs to be described how representative plant material was obtained for analyses. Fresh or frozen material is needed for enzyme isolation, so, how this was obtained if it is indicated in lines 354–358 that the plants were dried. Detailed description of extraction and activity measurement is necessary, as the reference to Dai et al. 2015 cannot be used, as this paper itself contains references to methods to older papers by the same scientific group.
Comments on the Quality of English LanguageThe English could be improved to more clearly express the research.
Author Response
Reviewer #2 (Comments and Suggestions for Authors)
We sincerely appreciate the substantial time and effort you have dedicated to reviewing our manuscript and offering such comprehensive, insightful, and constructive criticisms. Your feedback has accurately identified several critical issues in our original submission, including ambiguities in the methodology, limitations in the depth of discussion, and concerns regarding the rigor of conclusion derivation. We sincerely apologize for these shortcomings and have carefully revised and expanded the manuscript in response to each of your comments. We believe that the revised version demonstrates significant improvements in both scientific rigor and logical coherence. Below, we provide a point-by-point response to your specific comments.
Overall Comments: The main problems with this manuscript are related to the methodology section and discussion. The study is not replicable because essential information about the conditions and actions taken is missing. In addition, there are fundamental errors in the description of the methodology, which suggest that the authors did not perform some of the described actions. The discussion does not explain the main result highlighted – why does bacterial treatment alter Cd uptake?
Response: We fully accept your overall assessment of our manuscript. We acknowledge that the original submission lacked essential methodological details, which compromised the reproducibility of the study, and that the Discussion section did not adequately elaborate on the core findings. This shortcoming reflects an oversight in rigor and attention to detail during the drafting process, for which we sincerely apologize. In the revised manuscript, we have comprehensively rewritten the Discussion section and substantially improved the Materials and Methods section to fully address these concerns.
Comments 1: Title
Only the effect of the bacteria on Cd accumulation is emphasized, which is inherently contradictory, so the title does not match the content. In fact, the only clear result is the improvement in growth and changes in the antioxidant enzymatic system under the influence of the bacteria.
Response 1: Thank you for your valuable feedback. We agree that the original title was overly simplistic and potentially misleading. We have revised the title to better capture the complexity of the study and more accurately represent the research content. The new title is: “The effect of plant growth promoting rhizobacteria Bacillus thuringiensis LKT25 on cadmium accumulation and physiological responses in Solanum nigrum L.”
Comments 2: Abstract
Do not introduce abbreviations for terms not used more than three times within the abstract. ACC deaminase activity was not measured.
Response 2: We are sincerely grateful to the reviewers for their valuable comments.
Regarding abbreviations: In accordance with your suggestions, we have conducted a comprehensive review of all abbreviations used in the abstract. We have ensured that each abbreviation appears at least three times in the abstract before being introduced; otherwise, it has been replaced with its full form to improve clarity.
Regarding ACC deaminase, we sincerely appreciate your valuable correction on this matter. You are absolutely correct that the term "ACC deaminase activity" used in the original manuscript was imprecise and could potentially mislead readers. We sincerely apologize for this inaccuracy.
The experiment we performed was a qualitative screening based on the utilization of ACC as the sole nitrogen source in a selective medium. This assay was designed to assess the metabolic potential of the strain Bacillus thuringiensis LKT25 to utilize ACC, rather than to quantify enzymatic activity.
To fully address this issue and ensure accurate representation of our findings, we have made the following substantial revisions in the updated manuscript: We have systematically revised all instances of the inaccurate terminology, including "ACC deaminase activity" and similar expressions, throughout the entire manuscript (including the abstract, Materials and Methods, and Results sections).
In terms of result presentation, we have compiled the outcomes of this screening together with other plant growth-promoting traits (such as Siderophore production and Nitrogenase) into a newly created Table 1. This table provides a clear, qualitative summary of the strain's capabilities using a “+” or “-” format. Representative images supporting the screening results are now included in the Supplementary Materials.
Comments 3: Introduction
Mostly too general. Previous similar studies not analyzed in detail, as He et al. (2020) Ecotoxicol. Environ. Safety and Chi et al. (2023) J. Hazard. Mater. Therefore, it is not becoming clear why another study was necessary. The aim is not clearly formulated.
Several sentences describing particular facts have no references (lines 30–31; lines 56–57; lines 72–74)
Response 3: We have systematically revised and enhanced the Introduction section.
With regard to literature citations, we have added relevant authoritative references following all statements that lacked citation support, such as lines 30-31, 56-57, and 72-74, in order to strengthen the academic foundation of the discussion (lines 31, 79, 96 in the revised manuscript).
In terms of literature analysis, we have thoroughly examined and discussed the studies of He et al. (2020) and Chi et al. (2023) that were highlighted, along with other key related literature, and clearly delineated the differences between these prior studies and our research, particularly in terms of the bacterial strains employed, pollution concentration levels, and underlying mechanisms of action.
Moreover, at the conclusion of the Introduction, we have explicitly restated the specific research objectives of this study: to investigate the differential effects and underlying mechanisms of the cadmium-tolerant new strain LKT25, which possesses multiple plant growth-promoting traits, on the growth status, physiological responses, and cadmium accumulation characteristics of Solanum nigrum under varying cadmium stress conditions.
Comments 4: Results
Do not include facts from literature with references (lines 174–176; line 228).
In legends to figures 2 to 6, instead of "statistically significant differences", "significant correlations" is indicated.
Response 4: Thank you for your valuable reminder. We have relocated all literature citations that were inappropriately included in the Results section (e.g., lines 174-176 and line 228) to the Discussion section, where they are now appropriately used for comparative analysis. In the revised manuscript, the Results section exclusively presents our own experimental data in an objective and concise manner.
We sincerely apologize for the inappropriate use of statistical terminology in the figure captions. You are absolutely correct that "significant correlations" and "statistically significant differences" the latter being used for intergroup comparisons-represent distinct statistical concepts. We have conducted a comprehensive review and correction of all captions from Figure 2 to Figure 6, and have consistently revised erroneous expressions such as "significant correlations" to the accurate term "statistically significant differences".
Comments 5: Discussion
Too short and inconsistent with the results obtained.
The discussion places inappropriate emphasis, omitting conflicting information about the different effects of bacterial treatments on Cd accumulation. There is no explanation why at the lowest Cd concentration the bacterium promoted its accumulation in the shoot, at the medium concentration it had no effect, and at the highest concentration it reduced it. This is in no way related to the observed changes in enzyme activity, which were mostly positive.
Contradictions in results from the similar studies need to be discussed.
Response 5: This represents the core component of our revisions. We have comprehensively rewritten the Discussion section with a particular focus on addressing your key inquiry: why does the influence of bacteria on cadmium (Cd) accumulation differ across concentration levels? To address this question, we have formulated and elaborated on the following scientifically grounded hypotheses, supported by relevant literature:
At a Cd concentration of 5 mg/kg: The primary function of the bacteria is plant growth promotion. Through the secretion of indole-3-acetic acid (IAA), nitrogen fixation, ACC deaminase production, and siderophore synthesis, the bacteria significantly enhance plant biomass and root development. This leads to improved nutrient and mineral uptake including Cd resulting in an overall increase in Cd extraction.
At Cd concentrations of 25 and 50 mg/kg: The dominant function shifts toward detoxification. In this context, bacteria aid plants by promoting the compartmentalization of Cd within root tissues or its immobilization via binding to cell walls. This is achieved through the enhancement of antioxidant enzyme activities (e.g., SOD, POD, CAT, APX), the potential production of chelating compounds, and the induction of plant-related protein synthesis. These mechanisms actively restrict Cd translocation to the aboveground parts, thereby protecting vital metabolic processes such as photosynthesis. Consequently, although total plant biomass increases, the Cd concentration and content in the shoots either decrease or remain stable, reflecting a bacterial-mediated "intracellular detoxification" strategy.
In addition, we have integrated comparative analyses with findings from similar studies in the Discussion section, and provided plausible explanations for the observed differences.
Comments 6: Materials and methods
The weakest section, which raises doubts about the scientific integrity of the study.
In 4.2, it is claimed that IAA was dissolved in distilled water, but this compound is insoluble in water.
Lines 324–329 as if describe measurement of "ACC deaminase" activity. No activity was detected here. In fact, the provided reference is to composition of the original medium used by Dworkin and Foster. For assessment of bacteria, capable for using ACC as a nitrogen source through deaminase activity, ACC needs to be added to the medium. To measure deaminase activity itself, different methods can be used, but the simplest way is to record ACC breakdown rate using nonspecific ninhydrin assay.
In 4.4.1, no information is provided to understand how plant material was obtained, propagated, what were edaphic and other conditions (temperature, photon flux density of photosynthetically active radiation, photoperiod, relative humidity, fertilization, watering etc.) for plant cultivation, how long plants were cultivated etc.
In 4.4.2 it needs to be described how representative plant material was obtained for analyses. Fresh or frozen material is needed for enzyme isolation, so, how this was obtained if it is indicated in lines 354–358 that the plants were dried. Detailed description of extraction and activity measurement is necessary, as the reference to Dai et al. 2015 cannot be used, as this paper itself contains references to methods to older papers by the same scientific group.
Response 6: We sincerely regret any confusion or skepticism our Methods section may have caused. We would like to assure you that all described experiments were indeed conducted. However, our original description lacked professionalism and omitted critical details. We have now thoroughly revised and supplemented this section with comprehensive explanations:
IAA solubility: You are absolutely correct that IAA exhibits limited solubility in water. To address this, we first fully dissolved IAA in a small volume of 1 M NaOH solution, followed by dilution to the desired volume with distilled water. This detail has now been added to Section 4.2.
ACC deaminase: We sincerely apologize for the previous misrepresentation. Our experimental approach involved a selective medium assay using ACC as the sole nitrogen source to screen for bacterial strains capable of utilizing ACC. This method does not involve direct measurement of enzyme activity. The assay, referred to as "ACC utilization capacity screening," was based on the medium formulation described by Dworkin and Foster, which we have now properly cited. All references to "activity" have been removed accordingly.
Plant culture details (lines 423-444 in revised manuscript): We have now included all previously missing information, as detailed below: “This study used S. nigrum seeds collected from Hanzhong (33°7′50″N, 106°48′16″E) and was conducted at the Hanzhong laboratory of Shaanxi University of Technology. The soil, classified as brown-yellow soil (Xanthic Ali-Udic Cambisols of silt loam textural class according to WRB, 2015 classification), was sampled from the top layer (0-20 cm) on campus. It had a pH of 6.5, organic matter content of 26.8 g/kg, total nitrogen of 1.71 g/kg, available potassium of 17.3 mg/kg, available phosphorus of 16.8 mg/kg, and Cd content of 0.08 mg/kg. According to GB 15618–2018, the soil Cd level is Class І, representing a natural background concentration.
The pot experiments followed the method of Dai et al. [68], using soil spiked with CdCl₂·2.5H₂O in an analytical-grade reagent solution. The soil was equilibrated for two months before the experiment began. Treatments included one control group (CK, 0 mg/kg) and three Cd levels (5, 25, and 50 mg/kg), each with three replicates.
Each pot (25 cm × 15 cm) was filled with 2.5 kg of equilibrated soil. S. nigrum seeds were surface-sterilized with 0.1% HgCl₂ for 10 minutes before sowing. After germination, seedlings were grown under natural light. Four uniform seedlings (~5 cm tall) were transplanted into each pot and cultivated in a greenhouse environment characterized by a 16/8-hour photoperiod (day/night), a temperature regime of 25/20 °C (day/night). Plants were watered regularly with deionized water to maintain soil moisture at ~75% of field capacity.
During the 50-day cultivation period, the treatment group received root irrigation with 10 mL of bacterial suspension at 8-day intervals, for a total of three applications., while the control group received sterile water. At maturity, all plants were carefully uprooted and growth parameters were measured.”
Enzyme activity determination: We have provided comprehensive details regarding the methodology:
Sampling procedure (lines 465-469 in the revised manuscript): Fresh tissue samples (1 g) from the roots, shoots of S. nigrum were homogenized in 5 mL of pre-cooled 50 mmol/L phosphate buffer (pH 7.8) on ice. The resulting homogenate was transferred to a volumetric flask and diluted to a final volume of 10 mL. After centrifugation at 12,000 × g for 20 minutes at 4 °C, the supernatant was collected and used as the enzyme extract.
Comments on the Quality of English Language
The English could be improved to more clearly express the research.
Response: We have commissioned a professional English editing service (or collaborated with native English-speaking colleagues) to conduct a comprehensive language polishing of the entire manuscript, ensuring clarity, accuracy of expression, and adherence to academic writing standards. A certificate of language editing is enclosed with this submission.

Reviewer 3 Report
Comments and Suggestions for Authors
The manuscript describes the benefits of inoculating the plant growth-promoting rhizobacteria Bacillus thuringiensis LKT25 in the rhizosphere of the Solanum nigrum for the removal of Cd at different concentrations. The plant growth-promoting properties, such as phytohormone production, nitrogen fixation, and siderophore synthesis, of B. thuringiensis LKT25 were evaluated. Bacterial inoculation enhances S. nigrum growth by increasing biomass and chlorophyll production, as well as enhancing antioxidant activity. Finally, the Cd removal by S. nigrum was significantly improved, mainly at the concentration of 5 mg/L. Overall, the research described in the manuscript is interesting and pertinent.
The authors must address the following commentaries:
Line 44. The authors describe that Solanum nigrum L is a Cd hyperaccumulator plant. Due to this, it is necessary to complement this with information about the levels of Cd accumulation in plant tissues to be considered as a Hyperaccumulator plant, as well as the Cd bioaccumulation ranges reported for S. nigrum.
Line 56. Define the acronym “PGPR” for the first time used in the main text of the manuscript.
Line 75. Eliminate extra space in “Cd- tolerant”.
Lines 109, 131, 178, 188, 202, 203, 206, 230, and 233. Use “0, 5, 25, and 50 mg/kg” instead “0 mg/kg, 5 mg/kg, 25 mg/kg, and 50 mg/kg”.
Line 120. Eliminate period before “(“.
Lines 128, 144, 172, 197, 225, and 239. Check the asterisk indicator for p values, it could be “*, **, and ***”
Line 138. Use “25 and 50 mg/kg” instead “25 mg/kg and 50 mg/kg”.
Line 140. Eliminate period before “(“.
Line 149. Add a space in “1(Table 1)”.
Line 170. Add a period in “Table 1”.
Line 217. Review the space format.
Line 338, 373. Use italics for scientific names.
Author Response
Reviewer #3 (Comments and Suggestions for Authors)
We express our sincere gratitude for your dedicated time and substantial effort in reviewing our manuscript. Your insightful and constructive feedback is of great value to us. These comments have played a crucial role in improving the quality of our research work. In accordance with your suggestions, we have carried out a meticulous revision of the manuscript. The following is a point-by-point response to each of your specific comments, accompanied by a detailed account of the corresponding revisions. This allows you to clearly understand how we have addressed your concerns and enhanced the manuscript.
Comment 1: Line 44. The authors describe that Solanum nigrum L is a Cd hyperaccumulator plant. Due to this, it is necessary to complement this with information about the levels of Cd accumulation in plant tissues to be considered as a Hyperaccumulator plant, as well as the Cd bioaccumulation ranges reported for S. nigrum.
Response 1: We sincerely appreciate your invaluable suggestion. Your point is indeed well founded. When designating a plant as a hyperaccumulator, it is essential to present quantitative criteria for its accumulation ability. In accordance with your recommendation, we have incorporated the general critical value standards for hyperaccumulator plants and the specific accumulation data of Solanum nigrum in the introduction section (The revised manuscript can be found on line52-64).
In the revised manuscript, the following content was incorporated: “Hyperaccumulator plants, such as Solanum nigrum L, which is defined as a plant capable of accumulating more than 100 mg/kg Cd in its above-ground tissues under natural conditions, while maintaining a translocation factor (shoot-to-root ratio) greater than 1 [16]. However, different ecotypes of S. nigrum demonstrate considerable differences in their capacity to accumulate Cd under soil cultivation conditions. For example, the ecotype originating from the mountainous regions of Linhai City, Zhejiang Province, China, exhibited root and stem Cd concentrations ranging from 8.93 to 58.15 mg kg⁻¹ when exposed to soil Cd concentrations of 25–100 mg/kg [17]. In contrast, the population transplanted from Lizi Park in Nanshan District, Shenzhen City, Guangdong Province, displayed root and stem Cd concentrations of 122.7 mg/kg and 82.6 mg/kg, respectively, under a soil Cd concentration of 50 mg/kg [17]. Moreover, the Korean ecotype collected from Daegu, South Korea, showed Cd concentrations in roots, stems, and leaves ranging from 120.49 to 3162.83 mg/kg under soil Cd levels of 10–80 mg/kg [18].”
Comment 2: Line 56. Define the acronym “PGPR” for the first time used in the main text of the manuscript.
Response 2: We sincerely appreciate your bringing this oversight to our attention. We concur that when an abbreviation is employed for the initial time, the full name ought to be furnished. Consequently, we have duly made the requisite corrections.
In the original text, at the first occurrence of “PGPR”, it has been revised to “plant growth-promoting rhizobacteria (PGPR)”. The revised manuscript can be found on line 75.
Comment 3: Line 75. Eliminate extra space in “Cd- tolerant”.
Response 3: I sincerely appreciate your meticulous proofreading. We have now eliminated the extra space in "Cd-tolerant". The revised manuscript can be found on line 97.
Comment 4: Lines 109, 131, 178, 188, 202, 203, 206, 230, and 233. Use “0, 5, 25, and 50 mg/kg” instead “0 mg/kg, 5 mg/kg, 25 mg/kg, and 50 mg/kg”.
Response 4: Thank you for pointing out the error. We have systematically adjusted the listing format of concentrations in all the above - mentioned lines to the more concise form "0, 5, 25, and 50 mg/kg". For specific details, please refer to lines 141, 162, 204, 213, 227, 228, 231, 255, and 257 of the revised manuscript.
Comment 5: Line 120. Eliminate period before “(“.
Response 5: We sincerely appreciate you bringing this punctuation error to our attention. The period preceding the bracket in line 120 has been removed, as detailed in the revised manuscript on line 152.
Comment 6: Lines 128, 144, 172, 197, 225, and 239. Check the asterisk indicator for p values, it could be “*, **, and ***”
Response 6: Thank you for your reminder. We have meticulously reviewed all the aforementioned rows and their corresponding significance markers in the figures, and have standardized them to the format of "*" and "**" to denote significance levels of p < 0.05 and p < 0.01, respectively. Please refer to the revised manuscript on lines 159, 175, 201, 223, 251, and 265 for details.
Comment 7: Line 138. Use “25 and 50 mg/kg” instead “25 mg/kg and 50 mg/kg”.
Response 7: In accordance with your recommendation, the expression on line 138 has been revised to "25 and 50 mg/kg." Please refer to the modified manuscript on line 168 for details.
Comment 8: Line 140. Eliminate period before “(“.
Response 8: The period preceding the brackets in line 140 has been removed. The revised manuscript can be found on line 171.
Comment 9: Line 149. Add a space in “1(Table 1)”.
Response 9: A space has been added here, resulting in the modified format "1 (Table 1)." The revised manuscript can be found on line 180.
Comment 10: Line 170. Add a period in “Table 1”.
Response 10: A period has been added at the end of "Table 1". The revised manuscript can be found on line 200.
Comment 11: Line 217. Review the space format.
Response 11: We sincerely appreciate your attention to the formatting issue. We have thoroughly reviewed and rectified the improper spacing in line 217 of the original manuscript (e.g., ensuring single-space formatting between words). The revised manuscript can be found on line 242.
Comment 12: Line 338, 373. Use italics for scientific names.
Response 12: The species names in Lines 338 and 373 have been formatted in italics (e.g., Solanum nigrum, Bacillus thuringiensis) in accordance with biological nomenclature conventions. Please refer to the modified manuscript on line 413 and 464 for details.

Round 2
Reviewer 1 Report
Comments and Suggestions for Authors
The revised manuscript demonstrates a clear improvement, and I very much appreciate the authors’ efforts in addressing the previous suggestions as well as the carefully expanded discussion. The manuscript is now more coherent and of higher scientific quality.
I would, however, kindly recommend a few additional adjustments that could further improve the readability and consistency of the presentation:
1) Figures: The font size in figure legends and axis labels is too small to be easily readable (Figures 3–8). Adjusting it to at least 10–12 pt, as in Figure 2, would improve clarity.
2) Supplementary material: In Figure A2, the description refers to six strains, while the figure shows only one, and I this is not CAS medium. In addition, Figure A3 seems to represent siderophore production and the decolorization of CAS medium around the colonies rather than ACC deaminase activity. Clarification would be helpful.
3) References: Line 328 and onwards: Please verify consistency with journal style. For example, citations should be given as [36, 37] rather than [36-37]. Also, in line 299, the reference “[3,3-40]” seems to be a typographical error and likely intended as “[3, 30–40]” (or the different, appropriate intended reference range).
4) Text corrections:
-
- Lines 392–396 could be rewritten in past tense. You may also consider condensed description, e.g.: “A standard stock solution of 1000 mg/L IAA was prepared and diluted to obtain 0, 2.5, … solutions, which were used to construct the standard curve.” Technical details concerning the use of beakers or flasks are in my opinion unnecessary.
- Line 424: please correct the formatting of geographical coordinates by removing spaces (e.g., 33°7’50’’N, 106°48’16’’E).
- Lines 440 and 469: remove the space before the °C symbol.
Overall, these are relatively minor issues, and once addressed, I believe the manuscript will be well prepared for publication.
Author Response
Reviewer #1 (Comments and Suggestions for Authors)
We sincerely appreciate your thorough review of our revised manuscript and the positive feedback provided. We are equally grateful for your insightful supplementary suggestions, which have significantly contributed to enhancing the clarity and professional quality of our work. In response to each of your comments, we have conducted a meticulous review and implemented the necessary revisions, as detailed below:
Comment 1: Figures: The font size in figure legends and axis labels is too small to be easily readable (Figures 3-8). Adjusting it to at least 10-12 pt, as in Figure 2, would improve clarity.
Response 1: We sincerely appreciate your insightful feedback. In response, we have systematically adjusted the font sizes in all figures (Figures 3 to 8) to ensure that the legends and axis labels meet the minimum requirement of 10 pt, thereby enhancing the clarity and readability of the visual presentation.
Comment 2: Supplementary material: In Figure A2, the description refers to six strains, while the figure shows only one, and I this is not CAS medium. In addition, Figure A3 seems to represent siderophore production and the decolorization of CAS medium around the colonies rather than ACC deaminase activity. Clarification would be helpful.
Response 2: We sincerely appreciate the reviewer's thorough evaluation of our manuscript. Your observation is entirely accurate, and we sincerely regret this oversight. During the preparation of the supplementary materials, an unintentional error occurred in which Figures A2 and A3 were misplaced, leading to substantial misinterpretation. We are truly grateful for your constructive feedback in identifying this issue.
The corrected Figure A2 corresponds to the siderophore production assay, which clearly illustrates the orange-colored halo formed around bacterial colonies on CAS medium, indicating siderophore activity.
The corrected Figure A3 represents the qualitative assessment of ACC deaminase activity, demonstrating visible colony growth on DF salt minimal medium where ACC serves as the sole nitrogen source. In accordance with this correction, we have updated the figure caption to: "Figure A3. Growth of strain LKT25 on DF salt minimal medium with ACC as the sole nitrogen source."
Comment 3: References: Line 328 and onwards: Please verify consistency with journal style. For example, citations should be given as [36, 37] rather than [36-37]. Also, in line 299, the reference “[3,3-40]” seems to be a typographical error and likely intended as “[3, 30-40]” (or the different, appropriate intended reference range).
Response 3: We sincerely appreciate your valuable guidance in standardizing the reference format. To ensure full compliance with the journal's formatting requirements, we have conducted a systematic review of all references throughout the manuscript.
All consecutively numbered citation formats (e.g., [36-37]) have been systematically revised to align with the journal-mandated citation style, which specifies that individual references should be listed separately as [36,37]. (The revised manuscript can be found on lines 31,67,268,274,279,289,294,302,308,327,328,330,332,335,337,348).
Furthermore, the typographical error previously cited as [3,3-40] on line 299 has been accurately corrected to the appropriate citation range [33-40].
Comment 4: Text corrections:
- Lines 392–396 could be rewritten in past tense. You may also consider condensed description, e.g.: “A standard stock solution of 1000 mg/L IAA was prepared and diluted to obtain 0, 2.5, … solutions, which were used to construct the standard curve.” Technical details concerning the use of beakers or flasks are in my opinion unnecessary.
- Line 424: please correct the formatting of geographical coordinates by removing spaces (e.g., 33°7’50’’N, 106°48’16’’E).
- Lines 440 and 469: remove the space before the °C symbol.
Response 4: We have fully incorporated all text modification suggestions you provided with careful attention to detail.
Lines 392-396: This section has been revised to consistently employ past tense and adopts the concise syntactic structure you recommended. All superfluous technical descriptions concerning beakers and conical flasks have been removed to enhance clarity and conciseness. The revised version now reads: " A 1000 mg/L IAA stock solution was prepared by dissolving IAA in 1 M NaOH and diluting with sterile water. Serial dilutions were made to obtain concentrations of 0, 2.5, 5, 7.5, 10, 12.5, and 15 mg/L for the standard curve." (The revised manuscript can be found on lines 392-394).
Line 424: The formatting of the geographic coordinates has been standardized by eliminating the incorrect spacing. (The revised manuscript can be found on line 422).
Lines 440 and 469: All unnecessary spaces preceding the temperature unit "°C" have been removed to ensure consistency and adherence to formatting standards. (The revised manuscript can be found on lines 439, 471).

Reviewer 2 Report
Comments and Suggestions for Authors
In legends to figures 2, 4 to 8 and Table 2, "significant correlations" are indicated. This needs to be changed to "statistically significant differences".
Some methodological details still does not match the description of the actions taken and indicates a possible source of error.
Lines 400–405 describe measurement of "ACC deaminase" activity. Still, it is not mentioned that ACC was added as a source of nitrogen. In reference [66], methane or ethane was used as hydrocarbon sources on a N-containing medium. Instead, for assessment of bacteria, capable for using ACC as a nitrogen source through deaminase activity, ACC needs to be added to the medium. To measure deaminase activity itself, different methods can be used, but the simplest way is to record ACC breakdown rate using nonspecific ninhydrin assay.
In 4.4.2, APX needs to be extracted separately adding ascorbic acid, as the enzyme is inactivated without it. However, this medium cannot be used for extraction of other antioxidative enzymes.
Comments on the Quality of English LanguageThe English could be improved to more clearly express the research.
Author Response
Reviewer #2 (Comments and Suggestions for Authors)
We sincerely extend our heartfelt gratitude for your meticulous and in-depth review of our manuscript. Your insights are of paramount significance, precisely identifying the lingering inaccuracies and deficiencies within our methodological description. We earnestly apologize for these oversights and have comprehensively revised the manuscript in strict accordance with each of your invaluable suggestions. Your input has substantially elevated the scientific rigor of our research endeavor. The following is a point-by-point response to your comments.
Comments 1: In legends to figures 2, 4 to 8 and Table 2, "significant correlations" are indicated. This needs to be changed to "statistically significant differences".
Response 1: We offer our sincere apologies for the inadequate nature of our previous revisions. Your observation is entirely on point: "significant correlations" and "statistically significant differences" employed for inter-group comparisons represent distinct and separate concepts.
We have conducted a comprehensive examination and rectification of the figure captions for Figures 2, 4-8, along with the annotations of Table 2. All instances of the inaccurate term "significant correlations" have been consistently and precisely amended to "statistically significant differences".
Comments 2: Lines 400–405 describe measurement of "ACC deaminase" activity. Still, it is not mentioned that ACC was added as a source of nitrogen. In reference [66], methane or ethane was used as hydrocarbon sources on a N-containing medium. Instead, for assessment of bacteria, capable for using ACC as a nitrogen source through deaminase activity, ACC needs to be added to the medium. To measure deaminase activity itself, different methods can be used, but the simplest way is to record ACC breakdown rate using nonspecific ninhydrin assay.
Response 2: We are sincerely grateful for your continued attention to this critical matter. We deeply regret the persistent ambiguities and inaccuracies in our methodological description. We fully acknowledge your observation regarding the inappropriateness of reference [66].
We have revised Section "4.2. ACC Utilization Capacity Screening" to accurately reflect the experimental procedures we performed.
Specifically, we conducted a qualitative screening to assess the "ability to utilize ACC as the sole nitrogen source," using the DF salt basal medium developed by Dworkin and Foster [66]. As you correctly pointed out, ACC was added as the sole nitrogen source to the medium at a final concentration of 3.0 mM. The bacterial strains were then inoculated onto this medium, and their growth was monitored after incubation to evaluate their capacity to utilize ACC [55]. (The revised manuscript can be found on lines 398-400).
In addition, we have revised the terminology from "ability to synthesize ACC deaminase" to "ability to utilize ACC" throughout the manuscript to eliminate any potential misinterpretation related to quantitative measurement. (The revised manuscript can be found on lines 123,284).
Comments 3: In 4.4.2, APX needs to be extracted separately adding ascorbic acid, as the enzyme is inactivated without it. However, this medium cannot be used for extraction of other antioxidative enzymes.
Response 3: Thank you for raising an important technical point. You are absolutely correct that the extraction of APX requires a protective buffer containing ascorbic acid to prevent inactivation, which may interfere with the measurement of other enzymes such as SOD and POD.
We apologize for this oversight and have added the following clarification in Section 4.4.2 " Determination of Antioxidant Enzyme Activities "(The revised manuscript can be found on lines 466-469).:
"Plant samples for APX activity should be extracted separately using a 50 mM phosphate buffer (pH 7.8) containing 5.0 mM ascorbic acid. Fresh weight samples are ground on ice and centrifuged [70]."
"Samples for SOD, POD, and CAT assays should be extracted using the same buffer without ascorbic acid."
Comments on the Quality of English Language
The English could be improved to more clearly express the research.
Response: Thank you for your careful review and insightful comments. We have revised the manuscript accordingly to improve the clarity and flow of the English language. We believe the paper is much improved.
